# RegionUDF: Region-Aware Unsigned Distance Fields for Surface Reconstruction from Point Clouds

## Abstract

Distance fields offer a powerful representation for continuous geometry, yet current learning-based neural unsigned distance fields (UDFs) remain limited in their ability to capture data patterns and generalize to real-world open surfaces. Point-Based methods mitigate grid quantization errors but current work often oversmooth local details, as query features are obtained solely through interpolation of point-wise features which are aggregated over large receptive fields. To address this, we propose a *discriminative region representation* that fuses narrow neighborhood features with broader contextual point-wise features, and a *primitive-based region representation* that decomposes the query region into triplet-defined primitives, enabling the detailed encoding of local surface geometry and the clear distinction of multi-layer structures. Building on these designs, we propose *RegionUDF*, a region-aware UDF framework that achieves state-of-the-art open-surface reconstruction on both object- and room-level scenes, with additional validation on watertight shapes. Extensive experiments on synthetic and real-world datasets demonstrate superior accuracy and robust cross-domain generalization. Our source code will be available at *[no-name-for-blind-review]*.

## 1 Introduction

Implicit geometric representations have become a predominant paradigm in 3D vision, computer graphics, and robotics (Park et al., 2019; Huang et al., 2022; Mescheder et al., 2019; Chibane et al., 2020b; Yang et al., 2023). Unlike explicit representations such as meshes or point clouds (Hanocka et al., 2020; Badki et al., 2020; Lin et al., 2020), implicit representations model surfaces as continuous isosurfaces in space, overcoming the resolution limits of discrete methods and enabling high-fidelity reconstruction. Among them, signed distance fields (SDFs) and unsigned distance fields (UDFs) are the most common. SDFs (Tang et al., 2021; Peng et al., 2020; Huang et al., 2023) encode signed distances to the nearest surface, where the sign indicates whether a point lies inside or outside a watertight object. UDFs (Ren et al., 2023; Fainstein et al., 2024; Ye et al., 2022), in contrast, record only distance magnitudes, allowing representation of more general, non-watertight shapes and making them particularly suitable for real-world surface reconstruction.

Recent advances have explored direct neural mappings from spatial coordinates to UDFs, either via scene-specific optimization (Zhou et al., 2022; 2023) or by defining new distance fields such as orthogonal Lu et al. (2024) and line-segment fields (Ren & Hou, 2025). Optimization-Based methods, however, are restricted to single scenes without addressing cross-scene generalization. In contrast, learning-based approaches (Chibane et al., 2020b; Ye et al., 2022; Ren et al., 2023) must construct query features from point-cloud inputs to generalize across scenes. Grid-Based trilinear interpolation remains common but introduces quantization errors and surface detail loss. Point-Based alternatives, though less studied for UDFs, have been explored for SDFs (Boulch & Marlet, 2022; Wang et al., 2023; Ranade et al., 2025). While they better preserve fine details, most rely only on point locations to refine interpolation, neglecting richer regional information; thus, neighborhood structure is underrepresented and point-wise features alone fail to capture fine-grained geometry.

Point-Wise features capture broad receptive fields but often oversmooth local details. In point-based methods, they encode regional characteristics over large spatial ranges, whereas narrow neighborhood

features we defined are confined to the immediate vicinity of a query point. These two levels of representation, reflecting "global" and "local" perspectives, are inherently complementary, and their integration produces a more compact and discriminative region feature, as illustrated in Fig. 1. Furthermore, since those points are often distributed across discrete surface patches, the neighborhood point cloud can be decomposed into finer primitives that are deliberately constructed to preserve the underlying surface structure. Theoretically, complex structures can be constructed through the aggregation of simple primitives. It enables a more detailed characterization of local geometry, while aggregating the primitives preserves information about the neighborhood as a whole.

In this work, we propose a *discriminative region representation* that models a query neighborhood by fusing broad contextual point-wise features with narrow region features extracted from its surrounding neighbors. Naturally, narrow region features can be obtained by treating the query neighbors as an independent point cloud, enabling the use of standard point cloud analysis techniques. This offers a principled alternative to weighted interpolation for encoding the local region relative to the query. We further propose a *primitive-based region representation*. The neighborhood is partitioned into multiple triplets, each of which uniquely defines a planar patch. Through spherical projection and angular sorting, primitives are constructed to preserve surface structure, thereby enabling clear discrimination of adjacent layers when the query point lies within a multi-layer structure. We first extract primitive features using the discriminative region representation with triplet points and point-wise features. Each primitive is then treated as a point, with its primitive feature serving as the corresponding point-wise feature. Finally, these primitives are aggregated to construct the query region feature.

Building on the above ideas, we develop a **Region**-Aware **U**nsigned **D**istance **F**ields framework, **RegionUDF**, and evaluate it on both synthetic and real-world datasets. We further assess its cross-domain generalization by varying datasets and scene scales. In addition, existing learning-based UDF studies have seldom conducted quantitative evaluations in room-level scenarios, typically providing only qualitative visual results. Yet, room-level scenarios are inherently more complex and variable, making them a more rigorous and informative benchmark for assessing learning capability. Therefore, we place particular emphasis on evaluation in room-level scenarios. Overall, our results demonstrate that RegionUDF consistently outperforms existing methods. Our main contributions are as follows:

- We design a discriminative region representation that incorporates narrow, fine-grained region features for more accurate modeling of the query neighborhood.

- We propose a primitive-based region representation that decomposes the region into primitives, enabling detailed encoding of local geometry and clear distinction of multi-layer structures.

- We validate our model on both synthetic and real-world datasets, demonstrating superior performance in both intra-domain and cross-domain evaluations. Comprehensive ablation studies further confirm the effectiveness of our proposed approaches.

## 2 RELATED WORK

Surface reconstruction has long been studied, with discrete representations (Schönberger et al., 2016; Schönberger & Frahm, 2016) limited by spatial resolution and memory. Implicit representations have thus become mainstream, spanning objects to large-scale environments. Classic methods such as Poisson reconstruction (Kazhdan & Hoppe, 2013; Kazhdan et al., 2006), radial basis functions (Carr et al., 2001), and moving least-squares surfaces (Guennebaud & Gross, 2007; Levin, 1998) rely on smoothness priors, whereas neural implicit approaches offer greater expressiveness and flexibility.

**Closed Surface.** In distance field modeling, SDFs parallel occupancy fields by mapping distances to occupancy probabilities. Early neural implicit models such as DeepSDF (Park et al., 2019) and ONet (Mescheder et al., 2019) predict signed distances or occupancy scores from global shape codes. To better capture local geometry, voxel-based methods (Peng et al., 2020; Tang et al., 2021; Chibane et al., 2020a) adopt voxel-based encodings, though interpolating latent features limits fine detail. Spline-based methods (Williams et al., 2022; Huang et al., 2022) address this via local patches. NKSR (Huang et al., 2023) further refines the SDF by incorporating hierarchical information to solve the global gradient optimization problem. Optimization-based approaches (Gropp et al., 2020; Sitzmann et al., 2020; Ma et al., 2021) instead solve the Eikonal equation with neural PDE solvers (Sirignano & Spiliopoulos, 2018). A shared limitation is the assumption of closed surfaces, restricting applicability to open surfaces in real scans.

**Open Surface.** To represent open surfaces, NDF (Chibane et al., 2020b) first introduced neural unsigned distance fields, regressing unsigned distances from spatial queries to the surface. Subsequent per-scene optimization methods refined this representation: CAP-UDF (Zhou et al., 2022) employs a Chamfer-based pull objective, DUDF (Fainstein et al., 2024) enforces differentiability via hyperbolic scaling, and LevelSetUDF (Zhou et al., 2023) stabilizes gradients through point-projection. Other variants like PDDF (Aumentado-Armstrong et al., 2022), NeuralODF (Houchens et al., 2022), and LineSeg (Ren & Hou, 2025), incorporate directional cues for direct mesh extraction. Specifically, UODF (Lu et al., 2024) defines the minimal unsigned distance along three orthogonal directions, enabling each spatial point to directly access its closest surface point and thereby achieve high-precision reconstruction without interpolation errors. While effective, these remain per-scene methods without cross-domain generalization. Learning-based UDFs are less explored: GIFS (Ye et al., 2022) predicts binary extension flags, NVF (Yang et al., 2023) regresses vectors to nearest surfaces, GeoUDF (Ren et al., 2023) upsamples input points and normals for interpolation, and SALS (Ren & Hou, 2025) learns line segment–surface relations beyond the UDF paradigm. Despite these advances, existing approaches overlook that constructing query features with an emphasis on narrow neighborhood features can yield more accurate representations.

## 3 METHOD

We propose a region-aware unsigned distance field framework that integrates point-wise and narrow region features of each query neighborhood. Point-Wise features capture broad contextual information but often oversmooth fine details, whereas narrow region features preserve local structures yet are sensitive to noise. Their integration yields complementary and more discriminative region representations. To further enhance local representation, we propose a primitive-based region formulation, decomposing the neighborhood into finer primitives for more detailed characterization.

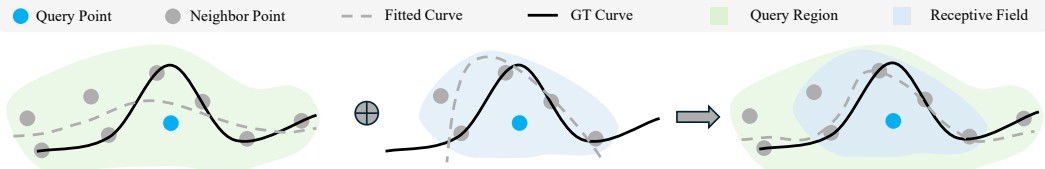

Figure 1: **Intuition of Discriminative Region Representation.** As an illustrative 2D example. The query identifies neighbors via KNN, defining the query region (blue). Through the point-wise features of these neighbors, the query also inherits a broader receptive field (green). Features restricted to the query region capture finer local detail but are sensitive to noise, whereas point-wise features alone oversmooth details. Their integration yields a more faithful/discriminative region representation.

### 3.1 DISCRIMINATIVE REGION REPRESENTATION

We consider narrow region features to form complementary, more *discriminative region representation* by fusing a narrow region feature $r_p$ with the point-wise feature $f_p$. Specifically, $f_p$ encodes multi-scale surface context from the input point cloud at each neighboring point. In parallel, $r_p$ encodes the spatial configuration of the region defined by the set $N_q$ of $K$ neighboring points surrounding $q$.

Treating the neighborhood as an independent point cloud with the query point $q$ as the origin of its coordinate system allows the direct application of point cloud analysis techniques. We follow the *pooling and propagation paradigm of PointNet++* (point-wise MLP $\rightarrow$ symmetric pooling $\rightarrow$ skip/concat). We formulate the whole distance learning process as:

$$r_p = \phi(p - q) \oplus (\frac{1}{K} \sum_p \phi(p - q)) | p \in N_q,$$
$$f_q = \delta(r_p \oplus f_p) | p \in N_q, \tag{1}$$
$$d_q = \varphi(f_q \oplus q)$$

where $\oplus$ denotes element-wise concatenation, $\phi$ denotes MLPs, $\delta$ means attention aggregation, $f_q$ represents the feature of the region relative to a query point $q$, and $\varphi$ is the regression MLP.

As shown in Eq. 1, we treat the set of neighboring points as a local region centered at the query point $q$. We apply a single layer of point-cloud abstraction followed by feature propagation to extract the narrow region feature $r_p$ as a narrow geometry representation relative to $q$ for each neighbor $p$. Concurrently, each neighbor is endowed with a point-wise feature $f_p$, which encodes a large receptive field and provides contextual information independent of $q$. By fusing the narrow region features $\{r_p\}$ with the query-independent features $\{f_p\}$, we construct a discriminative region representation that effectively characterizes the region relative to the query point.

## 3.2 PRIMITIVE-BASED REGION REPRESENTATION

However, because the region defined by the reference points often comprises multiple discrete surface patches, its holistic treatment in Eq. 1 may obscure fine-grained geometric details. To address this, we decompose the local region into a set of triplet primitives $\{S_j\}$, since three non-collinear points uniquely define a plane and complex shapes can then be represented as compositions of planar patches. For each $S_j$, we apply the paradigm of Eq. 1 to extract a primitive feature $f_{S_j}$. We then aggregate primitive features $\{f_{S_j}\}$ to form a richer representation of the region surrounding $q$.

First, we decompose the query-centered region into a collection of triplet primitives. Let $T = \{S_j\}_{j=1}^K$ denote the set of $K$ triplet primitives that together define the region surrounding the query point $q$. Accurately recovering surface connectivity from discrete reference points is challenging due to their sparse, unstructured nature. To mitigate this, we adopt an intuitive connectivity strategy based on angular proximity: we select the $K$ nearest neighbors of $q$, project them onto a virtual sphere centered at $q$, sort by spherical angles, and connect each three consecutive points to form a primitive. Through these operations, primitives are constructed to preserve surface structure, thereby enabling clear discrimination of adjacent layers when the query point lies within a multi-layer configuration.

For each triplet primitive $S = \{p_i \mid i = 0, 1, 2\}$ with the corresponding point-wise feature $f_{p_i}$, we could extract a primitive feature $f_S$ with Eq. 1 as following:

$$r_p = \phi(p - q) \oplus \left( \tfrac{1}{3} \sum_p \phi(p - q) \right) | p \in S,$$

$$f_S = \tfrac{1}{3} \sum_{p \in S} \phi(r_p \oplus f_p),$$

$$(2)$$

where $\phi$ is an MLP, $r_p$ is the hierarchical region feature at $p$, and $f_S$ is obtained via MLP and mean pooling. After computing each $f_S$, we aggregate these primitive features into a detailed region feature $f_q$ relative to the query $q$, also with paradigm defined by Eq. 1:

$$c_S = \tfrac{1}{3} \sum_{p \in S} (p - q),$$

$$r_S = \phi(c_S) \oplus \left( \tfrac{1}{K} \sum_S \phi(c_S - q) \right) | S \in T,$$

$$f_q = \delta\big(\{r_S \oplus f_S \mid S \in T\}\big),$$

$$d_q = \varphi(f_q \oplus q),$$

$$(3)$$

where $c_S$ denotes the center of the primitive $S$, $r_S$ means the hierarchical region feature at the primitive $S$. In practice, $\delta$ is the AttSet aggregation (Hu et al., 2020).

In practice, projecting reference points onto a sphere and sorting by angular coordinates via radix sort ensures minimal angular separation and coherent primitive formation. Given the simplicity of each triplet, a PointNet++-like structure suffices for feature extraction. Although each $f_S$ is relative to $q$, incorporating these region features yields a more discriminative representation of the region. For implement details, please refer to Appendix section A.2.

## 4 EXPERIMENTS

### 4.1 OVERVIEW

In this Section, we present a thorough evaluation of our method on both synthetic and real-world datasets. Specifically, we test on the watertight ShapeNet (Chang et al., 2015), the multi-layer open

surface "Car" subset of ShapeNet (Chang et al., 2015), on indoor scans from ScanNet (Dai et al., 2017) and Matterport3D (Chang et al., 2017). We also compare with the latest SALS on the ABC and non-manifold ABC datasets (Ren & Hou, 2025) it introduced. We further evaluate cross-domain performance across synthetic and real data and scene scales, supported by ablation studies on region-aware features and robustness tests under varying density and noise. For baseline reproduction details, see Appendix Section A.3.3. For cross-domain evaluation, see Appendix Section A.4.

## 4.2 WATERTIGHT SURFACES

Following established evaluation protocols, we assessed performance on watertight shapes from the ShapeNet 13-class dataset. For dataset details, please refer to Appendix Section A.3.1. For fairness, all experiments used 3K points as input. Given the large scale of the dataset, we directly adopted the comparative results reported in the GeoUDF (Ren et al., 2023) manuscript rather than reproducing them; consequently, the most recent SALS (Ren & Hou, 2025) is not included in this comparison. For evaluation metrics, please refer to Appendix Section A.2.3.

Table 1: **Watertight ShapeNet Comparison (Xu et al., 2019).** $CD_{L_1} \times 10^{-2}$, F-S. (%) with threshold 0.005 and 0.01. **Best** are in bold, and second-best are underlined. Note that Marching Cubes (MC) with a resolution of 128 was applied to all methods following the GeoUDF setting, while "+" indicates the higher resolution reported in their original papers.

| | Clean | | | | Noise (0.005) | | | |
|---|---|---|---|---|---|---|---|---|
| | $CD_{L_1} \downarrow$ | | F-S. $\uparrow$ | | $CD_{L_1} \downarrow$ | | F-S. $\uparrow$ | |
| | Mean | Median | $F1^{0.5\%}$ | $F1^{1\%}$ | Mean | Median | $F1^{0.5\%}$ | $F1^{1\%}$ |
| NDF (Chibane et al., 2020b) | 0.341 | 0.320 | 84.0 | 97.6 | 0.431 | 0.419 | 68.5 | 96.1 |
| GIFS(Ye et al., 2022) | 0.328 | 0.276 | 86.0 | 97.4 | 0.418 | 0.358 | 73.1 | 95.8 |
| GIFS$^+$(Ye et al., 2022) | 0.281 | 0.243 | 91.4 | 98.5 | 0.376 | 0.348 | 78.0 | 96.8 |
| GeoUDF (Ren et al., 2023) | 0.234 | 0.226 | 93.8 | 99.2 | 0.289 | 0.278 | 89.3 | 98.7 |
| **Ours** | **0.229** | **0.222** | **95.0** | **99.4** | **0.273** | **0.261** | **91.7** | **98.9** |

We compare our model with mainstream UDF methods for watertight shapes following the GeoUDF protocol. As shown in Table 1, our model consistently outperforms existing learning-based UDF approaches across all metrics.Our method achieves consistent improvements over GeoUDF, particularly in F-score, with gains of 1.2% on clean data and 2.4% on noisy data at the 0.5% threshold, indicating closer surface approximation and fewer reconstruction artifacts. As shown in Fig. 2, GIFS produces scaly textures, while GeoUDF introduces boundary gaps and fragmented details. In contrast, our method attains higher fidelity, accurately recovering thin structures and separating adjacent objects.

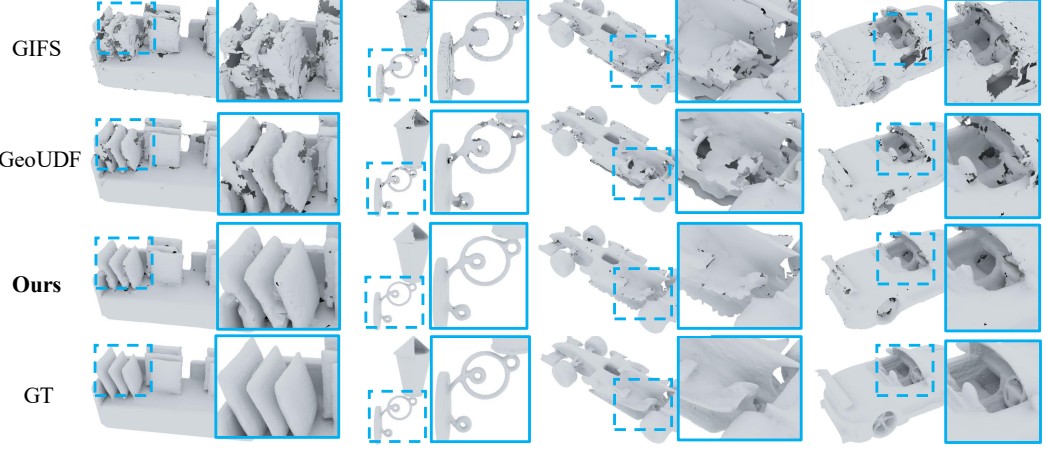

Figure 2: **ShapeNet (Chang et al., 2015) Visualization.** All methods are evaluated at a resolution of 128. Zoomed-in views highlight the regions with the most significant differences for comparison.

## 4.3 OPEN SURFACES

### 4.3.1 SURFACE RECONSTRUCTION OF SHAPES

To assess our method's capability on shapes with arbitrary topology, we follow prior work (Chibane et al., 2020b; Ye et al., 2022) and evaluate on the original "Car" category of ShapeNet, which features multi-layered and open-surface models. For evaluation metrics, please refer to Appendix Section A.2.3. Since GeoUDF (Ren et al., 2023) and SALS (Ren & Hou, 2025) do not provide training results on open surfaces, we retrain them on the non-watertight Car models using the default settings of their open source code and evaluate them under the same protocol. For SALS (Ren & Hou, 2025), resolution 256 exceeds a single RTX 3090's capacity, so we report results at 128; the default resolution is 256. We further compare against the recent learning-based open-surface reconstruction method SALS (Ren & Hou, 2025) on its proposed benchmark.

Table 2: **ShapeNet Cars (Chibane et al., 2020b) Comparison.** F-S. (%) with a distance threshold of 0.005 and 0.01, $d_C = CD_{L_2} \times 10^4$, and NC (%). Best are in bold, and second are underlined.

| | $d_C$ | | F-S. | | NC |
|---|---|---|---|---|---|
| | Mean↓ | Median↓ | F-S.$^{0.005}$ ↑ | F-S.$^{0.01}$ ↑ | ↑ |
| GeoUDF$^-$ (Ren et al., 2023) | 0.138 | 0.134 | 86.07 | 98.91 | 85.9 |
| NVF$^-$ (Yang et al., 2023) | 0.166 | 0.161 | 83.23 | 97.93 | 83.5 |
| SALS (Ren & Hou, 2025) | 0.179 | 0.175 | 83.05 | 97.49 | 77.4 |
| **Ours$^-$** | **0.135** | **0.131** | **86.42** | **98.93** | **86.4** |
| NDF (Chibane et al., 2020b) | 0.126 | 0.120 | 88.09 | **99.54** | - |
| NDF (Mesh) (Chibane et al., 2020b) | 0.202 | 0.193 | 77.40 | 97.97 | 79.1 |
| GIFS (Ye et al., 2022) | 0.128 | 0.123 | 88.05 | 99.31 | - |
| GeoUDF (Ren et al., 2023) | 0.120 | 0.114 | 89.23 | 99.29 | 86.6 |
| NVF (Yang et al., 2023) | 0.134 | 0.126 | 87.39 | 98.93 | 84.0 |
| **Ours** | **0.110** | **0.105** | **90.69** | 99.50 | **86.9** |

Table 3: **ABC and Non-Manifold ABC (Ren & Hou, 2025) Comparison.** F-S. (%) with a distance threshold of 0.005 and 0.01, $CD_{L_1} \times 10^{-2}, CD_{L_2} \times 10^{-6}$, and NC (%). Best are in bold, and second are underlined. Default Marching Cubes resolution is 128.

| | Method | CD | | F-S. | | NC |
|---|---|---|---|---|---|---|
| | | $CD_{L_1}$ | $CD_{L_2}$ | F-S.$^{0.005}$ ↑ | F-S.$^{0.01}$ ↑ | ↑ |
| ABC | NDF (Chibane et al., 2020b) | 0.324 | 15.4 | 86.31 | 98.66 | 80.6 |
| | GIFS (Ye et al., 2022) | 0.345 | 17.3 | 84.56 | 97.84 | 91.7 |
| | GeoUDF (Ren et al., 2023) | 0.257 | 9.02 | 92.88 | 99.78 | 97.2 |
| | SALS(Ren & Hou, 2025) | **0.251** | 8.87 | 92.66 | 99.73 | 97.3 |
| | **Ours** | 0.251 | **8.62** | **93.46** | **99.85** | **97.8** |
| Non-Manifold ABC | NDF (Chibane et al., 2020b) | 0.395 | 21.1 | 75.62 | 97.52 | 77.1 |
| | GIFS (Ye et al., 2022) | 0.412 | 22.6 | 73.78 | 97.71 | 89.9 |
| | GeoUDF (Ren et al., 2023) | 0.333 | 14.4 | 83.69 | 99.55 | 95.0 |
| | SALS (Ren & Hou, 2025) | 0.330 | 14.3 | 83.63 | 99.52 | 93.9 |
| | **Ours** | **0.320** | **13.2** | **85.65** | 99.74 | 96.8 |

For ShapeNet "Cars" benchmark, we compare RegionUDF against recent learning-based surface reconstruction methods, NDF (Chibane et al., 2020b), GIFS (Ye et al., 2022), GeoUDF (Ren et al., 2023) and SALS (Ren & Hou, 2025), on the ShapeNet Cars benchmark. Table 2 reports quantitative metrics, ours consistently achieves the best scores across most metrics and is comparable to the best normal consistency, indicating that ours produce fewer outlier artifacts.

For ABC and Non-Manifold ABC benchmark, in line with SALS default settings, we use 40k points without normals as input for each shape. *Please refer to Appendix Section A.3.3 for more details.* As shown in Table 3, our model achieves comparable CD performance while surpassing others on F-score and NC by about 1% on F-S$^{0.005}$ and NC, particularly on the non-manifold ABC benchmark, indicating fewer outlier artifacts and higher reconstruction quality.

As shown in Fig. 3, most methods struggle with complex non-manifold edges, an inherent limitation of UDFs. Our approach performs better on manifold regions, preserving sharp details along crisp

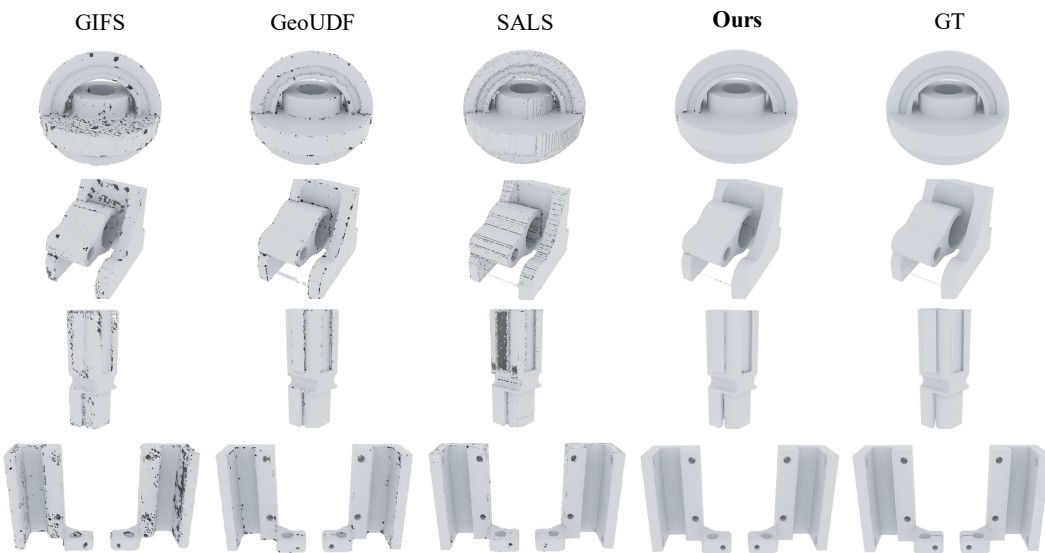

| GIFS | GeoUDF | SALS | **Ours** | GT |

Figure 3: **ABC and Non-Manifold ABC (Ren & Hou, 2025) Visualization.** All methods are evaluated at a resolution of 128 with non-manifold edges preserved and holes filled. The first row shows non-manifold objects, while the subsequent three rows depict the original ABC objects.

edges. In contrast, GIFS produces scaly surfaces, GeoUDF shows missing parts and artifacts, and SALS introduces excessive edge artifacts. This demonstrates that discriminative features help recover sharp details, counteracting the smoothing effects of point-wise features.

### 4.3.2 SURFACE RECONSTRUCTION OF ROOMS

To further assess our method on real-world scene scans, we evaluate it on two large-scale indoor datasets: ScanNet (Dai et al., 2017) and Matterport3D (Chang et al., 2017). For evaluation metrics, please refer to Appendix Section A.2.3. As no prior work reports room-level results on these benchmarks, we retrain them under default experimental settings reported in their papers and apply a consistent evaluation protocol for fair comparison. Due to SALS meshing limitations, experiments were conducted at two resolutions, with SALS results reported only at 128.

Table 4: **ScanNet and Matterport3D Comparison (Dai et al., 2017; Chang et al., 2017).** $CD_{L_1} \times 10^{-3}$, $CD_{L_2} \times 10^{-6}$, F-S. (%) with threshold 0.005, and NC (%). Best results are in bold, and second-best results are underlined. Default MC resolution is 256; "$^{-}$" denotes 128.

|  | ScanNet | | | | Matterport3D | | | |
|---|---|---|---|---|---|---|---|---|
|  | $CD_{L_1}\downarrow$ | $CD_{L_2}\downarrow$ | F-S. $\uparrow$ | NC$\uparrow$ | $CD_{L_1}\downarrow$ | $CD_{L_2}\downarrow$ | F-S.$\uparrow$ | NC$\uparrow$ |
| GeoUDF$^{-}$ (Ren et al., 2023) | 2.36 | 11.9 | 91.0 | 88.7 | 2.35 | 10.1 | 91.7 | 93.9 |
| NVF$^{-}$ (Yang et al., 2023) | 2.20 | 11.0 | 92.9 | 88.8 | 2.31 | 10.0 | 92.2 | 93.7 |
| SALS$^{-}$ (Ren & Hou, 2025) | **2.02** | 7.49 | **94.7** | 87.1 | 2.81 | 15.8 | 87.8 | 89.5 |
| **Ours$^{-}$** | **2.02** | **7.13** | 94.3 | **89.4** | **2.20** | **7.95** | **93.2** | **94.3** |
| NDF(Chibane et al., 2020b) | 2.31 | 8.60 | 93.3 | - | 2.54 | 9.81 | 91.8 | - |
| NDF (Mesh)(Chibane et al., 2020b) | 2.83 | 48.7 | 87.0 | 88.1 | 2.84 | 12.6 | 87.9 | 83.5 |
| GIFS (Ye et al., 2022) | 2.20 | 7.92 | 94.6 | 87.7 | 2.56 | 10.3 | 91.3 | 92.3 |
| GeoUDF (Ren et al., 2023) | 2.05 | 7.84 | 93.8 | 89.3 | 2.21 | 8.33 | 92.9 | 94.2 |
| NVF (Yang et al., 2023) | 2.03 | 11.7 | 94.6 | 88.9 | 2.83 | 11.9 | 89.5 | 93.6 |
| **Ours** | **1.86** | **6.03** | **95.7** | **90.1** | **2.09** | **7.29** | **94.1** | **95.5** |

As shown in Table 4, our method substantially outperforms existing approaches on real-world room-level surface reconstruction. NDF (Mesh) fails to produce coherent meshes on ScanNet. We achieve a **10%** improvement in $CD_{L_1}$ and nearly **20%** in $CD_{L_2}$ on ScanNet, and *5%* and *12%* improvements on Matterport3D, reflecting better recovery of fine details and closer alignment with ground-truth

surfaces. Additionally, NC increases by 1% and F-Score by nearly 2%, indicating improved geometric structure preservation. At resolution 128, our model perform a comparable performance as SALS on ScanNet while clearly outperforming it on Matterport3D, demonstrating the effectiveness of our region-aware features in capturing expressive details.

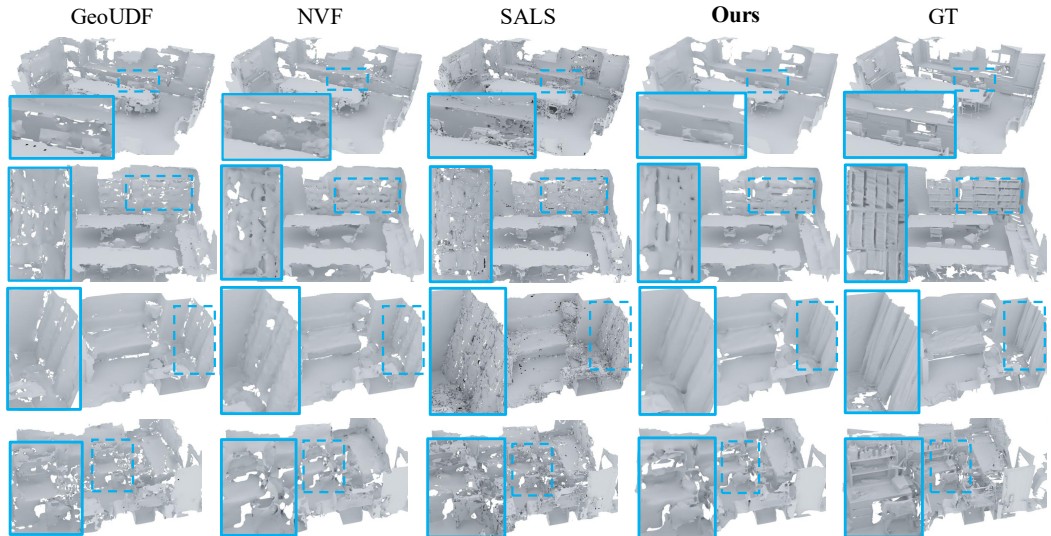

Figure 4: **ScanNet (Dai et al., 2017) Visualization.** All methods use a resolution of 128. Portions of the walls are removed to highlight interior details and more distinguishable structures.

Fig. 4 and Fig. 5 demonstrate the effectiveness of our model in room-level reconstruction. SALS often generates spurious structures and incomplete geometry, mainly due to the reduced number of input points and the limited capacity of its simple network. GeoUDF recovers the overall layout but produces noticeable distortions and fragmented artifacts near object boundaries, which cannot be corrected through standard post-processing because of incorrect local topology. NVF alleviates some of these issues by reducing missing regions, though it still struggles with complex multi-layered structures such as bookshelves, curtains, and clusters of pillows. In contrast, our method yields smooth and contiguous surfaces, reliably separates adjacent objects, and faithfully reconstructs fine details—including thin structures like curtains and pillowcases, while consistently preserving large planar surfaces such as beds and desks in densely cluttered environments.

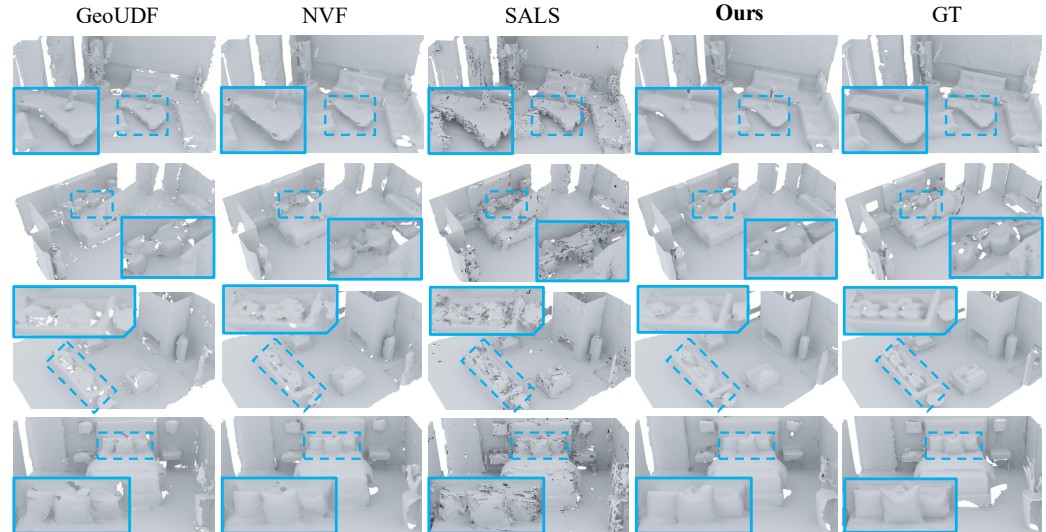

Figure 5: **Matterport3D (Chang et al., 2017) Visualization.** All methods use a resolution of 128. Portions of the walls are removed to highlight interior details and more distinguishable structures.

## 4.4 ABLATION STUDY

We conducted all ablation experiments on ScanNet, with default meshing resolution 256.

Table 5: **Method Ablation.** Report $CD_{L_1}$ and $CD_{L_2}$, F-S. with thresholds 0.005, 0.01, and NC.

| Methods | $CD_{L_1} \times 10^{-3}$ | | $CD_{L_2} \times 10^{-6}$ | | F-S. (%) | | NC (%) | Params. (M) |
|---|---|---|---|---|---|---|---|---|
| | Mean↓ | Median↓ | Mean↓ | Median↓ | F-S.$^{0.005}$ ↑ | F-S.$^{0.01}$ ↑ | ↑ | |
| $B_1$ | 1.92 | 1.91 | 6.33 | 6.02 | 95.3 | 99.5 | 89.0 | 11.32 |
| $B_2$ | 1.92 | 1.91 | 6.30 | 6.00 | 95.3 | 99.5 | 89.2 | 11.36 |
| $C_0$ | 2.10 | 2.07 | 8.20 | 7.73 | 93.7 | 99.1 | 88.6 | 11.37 |
| $C_1$ | 1.90 | 1.90 | 6.33 | 6.06 | 95.3 | 99.5 | 89.8 | 11.36 |
| $C_2$ | **1.86** | **1.87** | **6.03** | **5.82** | **95.7** | **99.6** | **90.1** | 11.37 |

**Method Design** All variants in Table 5 are controlled experiments sharing the exact same backbone (PointTransformer V2) and similar network capacity, with the only differences being the small modules designed to produce query features. We denote POCO-style baselines (represent positional encoding interpolation) include: (1) Direct UDF regression, which failed to converge after 300 epochs; (2) Incorporating query point locations as skip connections, similar to NDF and our method; (3) Replacing POCO's attention module with our AttSet module, based on (2), denoted as $B_1$ and $B_2$. We denote $C_0$ as the interpolation baseline (Eq. 4), $C_1$ as the architecture in Eq. 1 (Sec. 3.1), and $C_2$ as the primitive-based architecture (Eqs. 2–3). Notably, without the paradigm of Eq. 1, $C_2$ reduces to a weighted sum of point-wise features.

Specifically, $C_0 \rightarrow C_1$ isolates the effect of narrow region aggregation versus pure interpolation, $C_1 \rightarrow C_2$ isolates the contribution of primitive features, and $C_1$ versus $B_1/B_2$ isolates the impact of region-aware design versus positional-encoding enhancements. As shown in Table 5, the metrics for $C_0$, $C_1$, and $C_2$ exhibit a clear upward trend. $C_0$ serves as a baseline, whereas $C_1$ significantly surpasses it, demonstrating the benefit of narrow region features. $C_2$ further improves upon $C_1$, indicating that primitive-based aggregation effectively captures finer details. Comparisons between $B_1$ and $B_2$ reveal minimal impact from the AttSet module, while $C_1$ outperforms both, highlighting that narrow region features complement point-wise features to achieve superior representations.

Table 6: **Robustness Ablation.** Report $CD_{L_1}$ and $CD_{L_2}$, F-S. with thresholds 0.005, 0.01, and NC.

| Methods | Conditions | $CD_{L_1} \times 10^{-3}$ | | $CD_{L_2} \times 10^{-4}$ | | F-S. (%) | | NC (%) |
|---|---|---|---|---|---|---|---|---|
| | | Mean↓ | Median↓ | Mean↓ | Median↓ | F-S.$^{0.005}$ ↑ | F-S.$^{0.01}$ ↑ | ↑ |
| SALS | Clean | 2.02 | 2.02 | 0.075 | 0.069 | 94.7 | 99.3 | 87.1 |
| | Noise | 4.80 | 4.74 | 0.370 | 0.349 | 61.4 | 91.9 | 84.0 |
| | Noiser | 8.49 | 8.43 | 1.06 | 1.04 | 23.2 | 62.3 | 74.4 |
| | Sparse | 4.98 | 4.92 | 0.467 | 0.426 | 61.5 | 88.9 | 83.2 |
| GeoUDF | Clean | 2.05 | 2.04 | 0.078 | 0.075 | 93.8 | 99.0 | 89.3 |
| | Noise | 4.03 | 4.00 | 0.240 | 0.236 | 69.8 | 95.7 | 75.5 |
| | Noiser | 10.4 | 10.4 | 1.84 | 1.85 | 24.6 | 51.6 | 56.2 |
| | Sparse | 2.45 | 2.42 | 0.130 | 0.122 | 90.2 | 97.5 | 87.6 |
| NVF | Clean | 2.03 | 1.98 | 0.117 | 0.069 | 94.6 | 99.3 | 88.9 |
| | Noise | 2.88 | 2.84 | 0.171 | 0.142 | 87.2 | 97.6 | 85.6 |
| | Noiser | 5.20 | 5.23 | 0.503 | 0.469 | 62.0 | 86.9 | 78.6 |
| | Sparse | 2.77 | 2.73 | 0.199 | 0.175 | 87.2 | 96.1 | 85.1 |
| Ours | Clean | 1.86 | 1.87 | 0.060 | 0.058 | 95.7 | 99.6 | 90.1 |
| | Noise | 2.51 | 2.46 | 0.112 | 0.104 | 90.9 | 98.6 | 88.4 |
| | Noiser | 4.91 | 4.89 | 0.418 | 0.412 | 64.8 | 89.0 | 80.0 |
| | Sparse | 2.28 | 2.25 | 0.125 | 0.105 | 91.6 | 98.1 | 89.4 |

**Robustness** We evaluated robustness on ScanNet under three challenging conditions: (1) Gaussian noise with $\sigma = 0.005$ ("Noise"), (2) Gaussian noise with $\sigma = 0.025$ ("Noiser"), and (3) sparse input with 3,500 points, i.e., one-third of the original input ("Sparse").

As shown in Table 6, under the *Noise* setting, NVF exhibits a 41.9% degradation on $CD_{L_1}$, whereas our method achieves a noticeably smaller decline, outperforming NVF by **7%**. Under the *Sparse*

setting, NVF's degradation increases to 36.5%, while our method again demonstrates enhanced robustness with a **13.9%** smaller performance drop. In the *Noiser* setting, the degradation of our method becomes *comparable* to that of NVF. GeoUDF, however, displays substantial instability in noisy environments: its performance deteriorates by 96.5% under Noise and 407% under Noiser, whereas our method reduces these declines by **61.6%** and **243%**, respectively. Under sparse sampling, our degradation remains *close* to that of GeoUDF.

These results highlight that our formulation effectively exploits *independent neighborhood geometric cues* to suppress both noise and sparsity as long as local geometry remains partially preserved. When the local geometry is heavily destroyed by extreme noise (Noiser), our architecture naturally transitions to relying more on *broader-receptive-field point-wise features via attention*, yielding a controlled and acceptable degradation. Overall, the method demonstrates **balanced robustness across multiple distribution drifts**, maintaining strong performance in both noisy and sparse environments.

Table 7: **Primitive Construction.** Report $CD_{L_1}$ and $CD_{L_2}$, F-S. with 0.005, 0.01, and NC.

| | $CD_{L_1} \times 10^{-3}$ | | $CD_{L_2} \times 10^{-6}$ | | F-S. (%) | | NC (%) |
|---|---|---|---|---|---|---|---|
| | Mean↓ | Median↓ | Mean↓ | Median↓ | F-S.$^{0.005}$ ↑ | F-S.$^{0.01}$ ↑ | ↑ |
| Euclidean Distance-Based | 2.11 | 2.11 | 8.43 | 8.21 | 93.1 | 99.1 | 88.6 |
| Spherical Projection | **1.86** | **1.87** | **6.03** | **5.82** | **95.7** | **99.6** | **90.1** |

**Primitive Construction**    To demonstrate the effectiveness of our proposed primitive construction method, we compare it with a basic Euclidean distance-based partitioning method on the ScanNet.

As shown in the Table 7, Euclidean distance–based partitioning markedly degrades structural fidelity and thus harms reconstruction quality. Our spherical-projection grouping can produce ambiguity when query points lie outside a multilayer structure, but this drawback is mitigated because query points sampled between layers enable unambiguous separation of the two surfaces. During aggregation, attention further downweights ambiguous primitives and upweights those originating from between-layer queries, so the net effect of such ambiguity on the final reconstruction is small.

Table 8: **Primitive Construction.** Report $CD_{L_1}$ and $CD_{L_2}$, F-S. with 0.005, 0.01, and NC.

| | $CD_{L_1} \times 10^{-3}$ | | $CD_{L_2} \times 10^{-6}$ | | F-S. (%) | | NC (%) |
|---|---|---|---|---|---|---|---|
| | Mean↓ | Median↓ | Mean↓ | Median↓ | F-S.$^{0.005}$ ↑ | F-S.$^{0.01}$ ↑ | ↑ |
| Segment | 1.96 | 1.95 | 6.92 | 6.47 | 95.2 | 99.4 | 89.7 |
| Triplet Plane | **1.86** | **1.87** | **6.03** | **5.82** | **95.7** | **99.6** | **90.1** |
| Four-Point Patch | 2.06 | 2.04 | 7.54 | 7.05 | 94.3 | 99.3 | 88.3 |

**Primitive Type**    To demonstrate the effectiveness of our proposed triplet-plane primitive, we compare it with segment and 4-point patch on the ScanNet.

The choice of three points is motivated by geometric principles: in 3D space, the simplest non-degenerate local structure is defined by three non-collinear points, which uniquely determines a plane. Theoretically, simple 3D primitives provide one additional geometric degree of freedom while avoiding the excessive expressiveness of large patches. This intermediate complexity allows them to be learned reliably with low sample complexity and then composed to approximate rich surface structures. Results shown in Table 8 could support above claims.

## 5    CONCLUSION

In this paper, we present RegionUDF, a region-aware UDF framework that explicitly incorporates neighborhood information for each query region. We first design a discriminative region representation that fuses broad contextual point-wise features with narrow region features, providing complementary information. Building on this, we propose a primitive-based region representation that decomposes neighborhoods into triplet-defined primitives, enabling finer characterization of local geometry. Extensive experiments on synthetic and real-world benchmarks show that RegionUDF achieves superior reconstruction accuracy at both object and scene levels, with strong cross-domain generalization across diverse datasets and scales.

**Reproducibility Statement**  The formulas in the main text Section 3 directly correspond to our code. In Appendix Section A.2, we detail the backbone architecture, MLP configuration, and feature dimensions. We also provide training parameters and evaluation settings, including the meshing strategy, all based on official implementations. Furthermore, we describe the official baseline implementations used and how we retrained them in Appendix Section A.3.3.

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

# A APPENDIX

## A.1 PRELIMINARY

Naturally many works (Ye et al., 2022; Chibane et al., 2020b) follow the ONet (Mescheder et al., 2019) paradigm and the distance $d_q$ between query point $q$ and nearest surface is formulated as:

$$d_q = \varphi(\sum_{g \in N_q} w_{gq} f_g), \tag{4}$$

where $f_g$ denotes the grid feature of grid $g$ derived from point clouds $\boldsymbol{P}$ through the 3D Conv backbone, $w_{pq}$ is a scalar obtained from positions of the query point $q$ and the grid $g$, as trilinear interpolation, $N_q$ means the neighbors of the query point $q$, and $\varphi$ means a Regression Network, e.g. MLPs. It is well recognized that grid division imposes a fixed discretization, which inevitably introduces quantization errors and leads to the loss of fine-grained geometric details.

Recent point-based POCO (Boulch & Marlet, 2022) introduce attention-based interpolation to compute weights for feature aggregation:

$$d_q = \varphi(A(\phi(f_p|p))), \tag{5}$$

where $p$ is a neighbor of query $q$, $A$ means attentive pooling, and $\phi$ is a MLP to transform positions and point-wise features. However, these approaches largely ignore the inherent structural relationships within query neighborhoods, resulting a limited characterization of local regions. Point-Wise features capture broad contextual information over large receptive fields but often oversmooth local neighborhood details. In contrast, narrow region features faithfully describe neighborhood structures but may be prone to local extremes. These two levels of representation, global and local, are inherently complementary, and their integration yields a more compact and informative regional descriptor. Building on this, we propose a framework that explicitly combines point-wise features with narrow region representations to construct a discriminative representation of the query neighborhood.

## A.2 IMPLEMENTATION

We present the implementation of our method. For clarity, the extraction of point-wise features via PointTransformer V2 is omitted to highlight the core implementation (shown in Fig. 7).

**Implementation of Discriminative Region Representation.** In practical implementation, we uniformly sample 8 reference points per query and encode the fused geometric and contextual features $\{f_p, r_p\}$ through a lightweight two-layer MLP as shown in Eq.1. These per-point feature pairs are then aggregated using the attention-based AttSet mechanism (Hu et al., 2020) to form the final query feature $f_q$, which serves as input to the unsigned distance regressor. Since $r_p$ is computed from relative positions and $f_p$ is extracted by a translation-invariant backbone, the resulting $f_q$ inherits translation invariance, thereby ensuring robustness and stability in region representation. Additionally, the query point $q$ is incorporated via a skip connection to retain explicit positional information throughout the unsigned distance regression process.

**Implementation of Primitive-Based Region Representation.** We first reinforce the intuition behind primitive-based region description. As discussed in Section 3.1, narrow region features confined to the query region complement point-wise features, yielding more faithful and discriminative representations. A natural solution is to adopt finer-grained partitioning and feature learning, where the key challenge lies in constructing these fine-grained regions, termed primitives. While Euclidean distance-based neighbor selection is the simplest strategy, Fig. 6 illustrates that spherical projection more effectively aggregates patches on the same surface. The key lies in forming triplet primitives by angle-sorted adjacent points. While it introduces ambiguity in multi-layer structures, where patches may span different surfaces, such limitations are inherent to Euclidean distance–based method:

- When the query point is outside two layers, the primitives may treat them as a single surface, yet this limitation is not unique to our method and also appears in Euclidean distance–based partitioning.
- When the query point lies between layers, our formulation can effectively separate them, which Euclidean distance–based method cannot achieve.

- Moreover, during primitive aggregation, the attention mechanism operates over primitive centers, enabling the model to reinterpret ambiguous primitives as representing the "in-between" region of two layers. These primitives are then compensated by those constructed from query points located between the layers, which provide clear and consistent cues.

Thus, ambiguity may introduce minor artifacts, such as closer surfaces or a few unintended connections, but overall the method retains clear advantages over Euclidean distance–based partitioning.

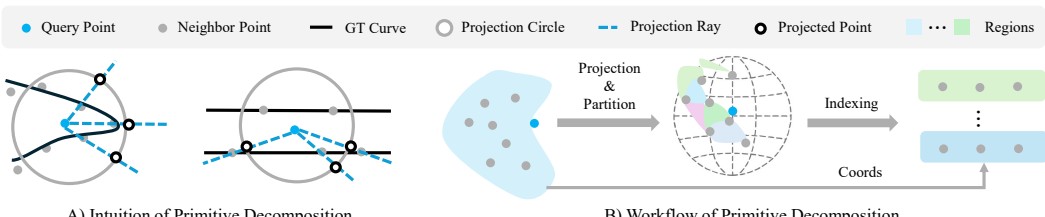

A) Intuition of Primitive Decomposition    B) Workflow of Primitive Decomposition

Figure 6: **Primitive Decomposition.** Subfigure A illustrates primitive decomposition in 2D. The projection method preserves sharp details and discrete patches by prioritizing surface proximity over Euclidean distance. Subfigure B shows that points are angle-sorted for primitive construction, while projection serves only for partitioning—the final regions remain defined in Euclidean space.

Next, we describe the network framework for primitive-based region representation. As illustrated in Fig. 7, the query region is first decomposed into individual primitives, each corresponding to a distinct subregion. These primitives, together with point-wise features, are input into the region feature constructor to learn primitive features. The primitives are then treated as points within the query region, with their features corresponding to the associated point-wise features, and re-input into the region feature constructor to produce the final query region feature. Note that, given the simplicity of triplet primitives, mean pooling is sufficient in the region feature constructor, whereas learning the final query region features requires attentive pooling.

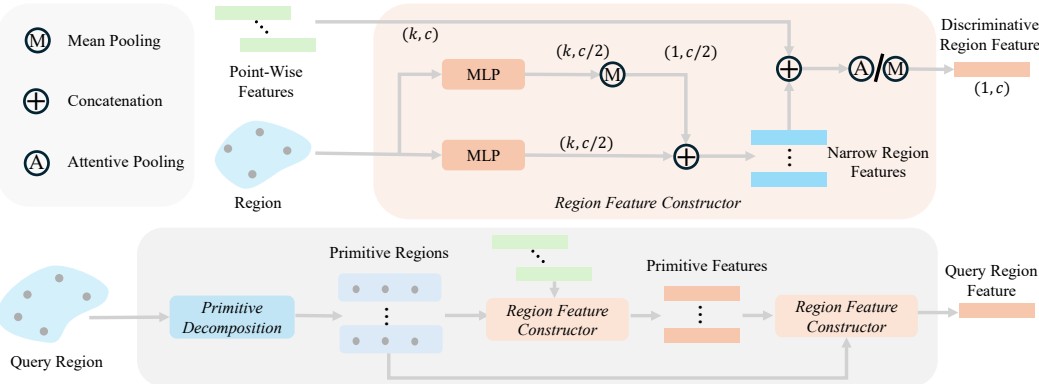

Figure 7: **Primitive-Based Region Description.** A query region is decomposed into primitives, each defined by its points and associated point-wise features. Primitive features are built via mean pooling in Region Feature Constructor, while final query region features need attention pooling. Primitive features serve as point-wise features, while primitives serve as points within the region.

**Loss Function**  Both the UDF value and UDF gradient are essential for optimization. The distances are optimized using $\ell_1$ loss, while the gradient of the field is used to compute the cosine similarity:

$$
\begin{aligned}
l_d &= \|d_{pd} - d_{gt}\|_1, \\
l_g &= 1 - |<\overrightarrow{g_{pd}} \cdot \overrightarrow{g_{gt}}>|,
\end{aligned}
\tag{6}
$$

where $l_d$ denotes the loss of query distances, $l_g$ means the similarity loss of gradient. Since the UDF is not differential at the iso-surface, the normal vector constraint on the surface is omitted. The loss

function is the combination of the two, as $l = \lambda_1 l_d^{off} + \frac{\lambda_1}{10} l_d^{on} + \lambda_2 l_g^{off}$. In our experiments, $\lambda_1, \lambda_2$ are hyper-parameters for weighting, $on$ means on-surface queries, $off$ means off-surface queries.

**Evaluation Setting**  During testing, we input sparse point clouds and adopt the meshing strategy introduced in MeshUDF (Guillard et al., 2022). During evaluation, we follow the protocol established in GIFS (Ye et al., 2022), because the MeshUDF meshing strategy can produce artifacts at low resolutions, especially when query points lie between multiple layers, we adopt the more robust, though less efficient, GeoUDF (Ren et al., 2023) meshing strategy at a resolution of 128. As evaluation metrics, including Chamfer Distance, F-Score, and Normal Consistency, are computed by uniformly sampling points from both the predicted meshes (or dense point clouds) and the ground truth meshes.

**Params. Setting**  Our framework employs PointTransformer V2 (Wu et al., 2022) as the backbone with default settings except initial grid size and downsample ratio. Two configurations are used: for simple watertight shapes with 3k points, the initial grid size is 0.015 with a downsample ratio of 2 per layer; for complex open-surface reconstruction, the initial grid size is 0.01 with a downsample ratio of 2.5. Each layer uses 16 neighbors to ensure a large receptive field, and the number of nearest neighbors $K$ per query point is fixed at 8 across all experiments.

The point-wise features are 48-dimensional vectors derived from the input point cloud through backbone. As MLP in Region Feature Constructor, we set it with one linear layer and one ReLU layer, transforming a 4-dimensional position vector (with the length of the vector) to a 24-dimensional feature vector. Note that the second optional mean pooling in the region feature constructor comprises two Linear layers, two ReLU layers, followed by a mean pooling layer. The attentive pooling module is formulated as: $f = softmax(linear(f)) * f$. The final regression network is 4-layers MLP with channels [51,256,32,32,1]. Note that, aside from the point-based backbone which uses batch normalization for feature extraction, our core implementation does not rely on BN.

### A.2.1 TRAINING SETTING

We adopt the ADAMW optimizer with default parameters, initializing the learning rate at $10^{-3}$. A warm-up phase of 2000 steps is applied at the beginning of training. Furthermore, the learning rate is decayed to $30\%$ of its current value at epochs 30, 70, 110, and 200 for all datasets. For dataset-related settings refer to the Experiment section 4. For loss, $\lambda_1 = 1, \lambda_2 = 1e^{-3}$. For Result 2, we trained on 2 RTX3090 GPUs with batch 4 for 56 hours, with other two datatsets results 4 trained on 2 RTX3090 GPUs with batch 4 for 24 hours. The CPU core is Intel Xeon Platinum 8383C CPU @ 2.70 GHz.

### A.2.2 MESHING STRATEGY

At the zero level set of an unsigned distance field (UDF), the derivative is undefined, leading to unstable gradients as the predicted values approach this boundary. This instability complicates the simultaneous optimization of distance and gradient losses (Eq. 6) in the vicinity of the zero level set. Mesh extraction methods for UDFs, such as the Marching Cubes adaptation used in MeshUDF (Guillard et al., 2022), depend on reliable gradient computations to assign relative signs via gradient dot products. When gradients fluctuate excessively near surface, mesh extraction suffers from missing or erroneous edge intersections, producing holes and spurious artifacts. As a result, *the meshing strategies can amplify gradient extremes, yielding visible gaps or distorted geometry*.

In our implementation, we adopt the MeshUDF pipeline to extract iso-surfaces from the predicted UDF. For object-level reconstruction, we use a voxel grid of resolution 128 and discard vertices whose distance estimates exceed $\frac{1}{3}$ or $\frac{1}{1.5}$ of the voxel size from the zero level set. For room-level reconstruction, we employ resolutions of 128 and 256. Since MeshUDF often introduces artifacts at low resolutions—particularly when query segments are inclined to the plane without intersection—we instead use GeoUDF at resolution 128 despite its higher cost.

### A.2.3 EVALUATION METRICS

Evaluation metrics included Chamfer Distance (CD) and F-Score (F-S.) and Normal Consistency (NC). For object-level evaluation, they are computed by sampling 100k points from both the reconstructed surfaces and the ground truth. For room-level evaluation, they are computed by sampling 500k

points from both the reconstructed surfaces and 100k points from the ground truth. **All objects are normalized to the unit cube for comparison.**

### A.3 EXPERIMENT SETTINGS

#### A.3.1 DATASET INTRODUCTION

We include 1 watertight and 4 open-surface reconstruction benchmarks as follows:

**ShapeNet** (Chang et al., 2015) is an object-level 13 classes watertight dataset processed by the DISN (Xu et al., 2019), with train/val/test split according to 3D-R2N2 (Choy et al., 2016).

**ShapeNet Cars** (Chang et al., 2015) is an object-level dataset of non-watertight car models featuring complex internal architectures of ShapeNet core dataset. Following the original NDF split (Chibane et al., 2020b), we use 5,249 scans for training, 749 for validation, and 1,499 for testing, while employing 10k points as input rather than occupancy data.

**ABC & Non-Manifold ABC** (Ren & Hou, 2025) is an object-level open-surface reconstruction benchmark introduced by SALS (Ren & Hou, 2025). Following its protocol, the model is trained on 100 shapes from Thingi10K (Zhou & Jacobson, 2016) and evaluated on 50 shapes from ABC (Koch et al., 2019), referred to as manifold ABC, as well as on 50 randomly selected shape pairs from ABC that are intersected to construct non-manifold ABC.

**ScanNet** (Dai et al., 2017) comprises 1,513 RGB-D scans of indoor scenes with complex topology and noisy, open surfaces. We adopt the standard split of 1,201 training and 312 validation scans; since test-set GT meshes are unavailable, all quantitative results are performed on the validation set.

**Matterport3D** (Chang et al., 2017) contains 10,800 panoramic RGB-D views derived from 194,400 images across 90 building-scale scenes. We follow the published partitioning of 61 scenes for training, 11 for validation, and 18 for testing. Each building-scale scene is further divided into room-level segments by Officials, and our experiments are conducted on the segmented dataset.

#### A.3.2 TRAINING DATA GENERATION

For both object-level and indoor datasets, we adopt the preprocessing pipeline like NDF (Chibane et al., 2020b), normalizing each ground-truth mesh to fit within **the unit cube**. During training, we sample query points both on and off the surface: specifically, we generate 10K on-surface points and 100K off-surface points per scene following the NDF sampling strategy. For each query point, we compute the nearest surface point to derive the corresponding distance and directional vector. Notably, for watertight shapes, we follow the convention of using 3k input points. For the ABC benchmark introduced by SALS, 40k points are sampled to maintain consistency.

#### A.3.3 REPRODUCE BASELINES

Since many learning-based UDF approaches do not provide published results on room-scale scans, we reproduce their scene-level performance by retraining each method using the default settings reported in their papers and evaluating under a consistent protocol. All methods are compared using the same evaluation pipeline; specific implementation details are provided below.

NDF (Chibane et al., 2020b): Official implementation. We retrain NDF using its default settings and employ MeshUDF (Guillard et al., 2022) as the meshing strategy to extract meshes.

GIFS (Ye et al., 2022): Official implementation. We retrain GIFS with default settings. For mesh extraction, we set the resolution to 128 and 256 instead of the original 160.

NVF (Yang et al., 2023): Official implementation. We retrain NVF using its default settings. As the journal version code is not yet available, we evaluate only the conference version.

GeoUDF (Ren et al., 2023): Official implementation. We retrain GeoUDF with default settings and align the number of input points in its upsampling module to match our experiments.

SALS (Ren & Hou, 2025): Official implementation. We retrain SALS under default settings. *This method introduces the ABC and non-manifold ABC benchmarks. Since the authors did not release*

*their pretrained model or detailed dataset, we reproduce the experiments with their offical codes under a consistent setting and report our reproduced results.*

POCO (Boulch & Marlet, 2022): Official implementation. We directly adopt the query feature construction code to implement $B_1$ and $B_2$ in Section 4.4.

### A.3.4 VISUALIZATION ANALYSIS

For all figures, we use Blender 4.5.3 for rendering. Meshes are post-processed in MeshLab with three steps: Remove Non-Manifold Edges, Fill Holes, and HC Laplacian Smooth. For GeoUDF and SALS, which produce many non-manifold structures, these steps have limited effect, so we additionally apply the Smooth with Angle filter (30° threshold). *Overlapping and non-manifold structures can yield erroneous normals in Blender, producing black shadows that are artifacts rather than missing geometry, with negligible impact on distance-based metrics such as CD and F-Score.* The scaly texture observed in GIFS is further accentuated by the rendering method.

### A.4 MORE RESULTS

### A.4.1 CROSS-DOMAIN RECONSTRUCTION

Beyond intra-domain reconstruction, we evaluate cross-domain generalization by training on one dataset and testing on two others. We consider shape-to-scene, scene-to-scene, and scene-to-shape transfers to quantify performance. These experiments demonstrate the robustness and adaptability of our method across diverse datasets and varying scales of reconstruction scenarios.

Table 9: **Cross-Domain Evaluation Results.** $CD_{L_1} \times 10^{-3}$, $CD_{L_2} \times 10^{-6}$, F-S. (%) with a threshold of 0.01, and NC (%). Best results are in bold, and second-best results are underlined.

| Trained on | $CD_{L_1}\downarrow$ | $CD_{L_2}\downarrow$ | F-S. ↑ | NC ↑ | $CD_{L_1}\downarrow$ | $CD_{L_2}\downarrow$ | F-S.↑ | NC ↑ |
|---|---|---|---|---|---|---|---|---|
| ShapeNet Cars | | Tested on ScanNet | | | | Tested on Matterport3D | | |
| NDF (Chibane et al., 2020b) | 2.64 | 12.6 | 98.0 | - | 2.80 | 11.8 | 98.9 | - |
| NDF (Mesh) (Chibane et al., 2020b) | 2.76 | 15.0 | 98.3 | 78.0 | 3.32 | 26.2 | 97.7 | 81.6 |
| GIFS (Ye et al., 2022) | 2.50 | 11.7 | 98.2 | 87.3 | 2.79 | 12.6 | 98.7 | 92.4 |
| NVF (Yang et al., 2023) | **2.11** | 9.19 | **98.8** | 87.6 | 2.33 | 10.0 | 98.8 | 91.9 |
| GeoUDF (Ren et al., 2023) | 2.44 | 13.0 | 97.3 | 86.0 | 2.48 | 11.6 | 98.3 | 91.5 |
| **Ours** | **2.11** | **8.92** | **98.8** | **88.0** | **2.23** | **8.59** | **99.2** | **93.3** |
| Matterport3D | | Tested on ScanNet | | | | Tested on ShapeNet Cars | | |
| NDF (Chibane et al., 2020b) | 2.40 | 9.60 | 98.9 | - | 3.20 | 13.4 | 99.4 | - |
| NDF (Mesh) (Chibane et al., 2020b) | 2.80 | 15.6 | 97.1 | 88.2 | - | - | - | - |
| GIFS (Ye et al., 2022) | 2.31 | 9.09 | 98.9 | 87.8 | 3.44 | 16.4 | 98.2 | 83.0 |
| NVF (Yang et al., 2023) | 1.98 | 7.90 | 99.2 | 89.0 | 3.31 | 16.2 | 98.0 | 81.3 |
| GeoUDF (Ren et al., 2023) | 2.06 | 7.91 | 99.0 | 89.0 | 2.99 | 12.4 | 99.2 | **85.2** |
| **Ours** | **1.88** | **6.65** | **99.4** | **91.1** | **2.89** | **11.4** | **99.5** | 85.1 |
| ScanNet | | Tested on ShapeNet Cars | | | | Tested on Matterport3D | | |
| NDF (Chibane et al., 2020b) | 3.25 | 13.9 | 99.2 | - | 2.61 | 10.2 | 99.2 | - |
| NDF (Mesh) (Chibane et al., 2020b) | 3.76 | 130 | 98.4 | 74.4 | - | - | - | - |
| GIFS (Ye et al., 2022) | 3.49 | 17.4 | 97.7 | 81.6 | 2.68 | 11.5 | 98.7 | 91.1 |
| NVF (Yang et al., 2023) | 3.45 | 17.9 | 97.5 | 79.2 | 3.17 | 38.5 | 98.1 | 92.0 |
| GeoUDF (Ren et al., 2023) | 3.03 | 12.7 | 99.2 | 84.6 | 2.26 | 8.69 | 99.2 | 93.7 |
| **Ours** | **2.90** | **11.5** | **99.4** | **85.2** | **2.14** | **7.61** | **99.5** | **94.6** |

As shown in Table 9, NDF (Mesh) exhibits almost complete failure when generalizing from scene-level to shape-level reconstruction on Matterport3D, due to large errors in NDF predicted UDF gradients. By contrast, our method consistently outperforms existing approaches across almost all cross-domain scenarios, and matches their Normal Consistency when transferring from scene-level to shape-level. In the shape-to-scene setting, our model achieves performance comparable to NVF, while lower $CD_{L_2}$ and higher NC. Notably, our model outperforms all baselines across all metrics in the scene-to-scene transfer setting. They demonstrate that our reconstruction results more closely align with GT surfaces, and the larger $CD_{L_2}$ improvement indicates a reduction in outlier artifacts.

The visual results Fig. 8 further demonstrate that our method excels at recovering sharp and fine-grained details. At the scene-to-object level, it accurately reconstructs two closely positioned seats within a vehicle. At the object-to-scene level, it yields noticeably sharper and more coherent object boundaries. At the scene-to-scene level, it recovers challenging structures such as wall-mounted bed curtains and crisp window-sill edges—features that competing methods fail to preserve.

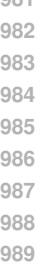
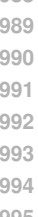
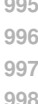

| GIFS | GeoUDF | NVF | **Ours** | GT |
| --- | --- | --- | --- | --- |

Figure 8: **Cross-domain evaluation results.** The first row shows scene-to-object transfer from Matterport3D to ShapeNet-Cars. The second and third rows illustrate object-to-scene transfer from ShapeNet-Cars to Matterport3D and ScanNet, respectively. The fourth row presents scene-to-scene transfer from ScanNet to Matterport3D. All views highlight the most distinctive regions.

### A.4.2 ABLATION STUDY

**Gradient Loss** To further illustrate the sensitivity of our method to gradient supervision, we conducted experiments on ScanNet, adjusting the coefficients $\lambda_2$. We observe that our method performs best with a gradient weight of $1e^{-3}$; however, further reduction or even the absence of gradient supervision results in only minor performance degradation. This indicates that incorporating region-aware features improves robustness to the gradient supervision term.

Table 10: **Gradient Weight Ablation.** $d_C = CD_{L_2} \times 10^6$, F-S. (%) with threshold 0.005, NC (%).

| Methods | $\lambda_2 = 0$ | | | $\lambda_2 = 1e^{-2}$ | | | $\lambda_2 = 1e^{-3}$ | | | $\lambda_2 = 1e^{-4}$ | | |
| --- | --- | --- | --- | --- | --- | --- | --- | --- | --- | --- | --- | --- |
| | $d_C \downarrow$ | F-S.↑ | NC↑ | $d_C \downarrow$ | F-S.↑ | NC↑ | $d_C \downarrow$ | F-S.↑ | NC↑ | $d_C \downarrow$ | F-S.↑ | NC↑ |
| $C_1$ | 7.56 | 94.9 | 89.4 | 9.52 | 92.7 | 88.0 | **6.33** | **95.3** | **89.8** | 7.71 | 94.0 | 88.7 |
| $C_2$ | 6.16 | 95.5 | 89.7 | 6.87 | 94.9 | 89.3 | **6.03** | **95.7** | **90.1** | 6.24 | 95.4 | 89.8 |

Our region-aware features offer stronger representational capacity than alternative designs, enabling the model to learn a more accurate distance field and thereby reducing reliance on gradient supervision. Since the gradient field is directly derived from the distance field, once the latter is well captured, gradients can be reliably obtained without heavy explicit supervision. In this way, a more accurate distance field naturally **reduces the marginal utility** of the gradient loss as shown in Table 10. $C_1$ exhibits relatively *weaker feature representation capabilities*, so it **relies more heavily** on gradient supervision to converge to a better distance field.

**KNN Ablation** Since we emphasize the complementarity of narrow region features with point-wise features that provide larger receptive fields, the effect of KNN size—which directly determines the query region—on narrow region features warrants ablation.

Table 11: **KNN Size.** Report $CD_{L_1}$ and $CD_{L_2}$, F-S. with thresholds 0.005, 0.01, and NC.

| K | $CD_{L_1} \times 10^{-3}$ | | $CD_{L_2} \times 10^{-6}$ | | F-S. (%) | | NC (%) |
|---|---|---|---|---|---|---|---|
| | Mean↓ | Median↓ | Mean↓ | Median↓ | F-S.$^{0.005}$ ↑ | F-S.$^{0.01}$ ↑ | ↑ |
| 4 | 2.00 | 2.00 | 7.40 | 7.06 | 94.2 | 99.4 | 88.6 |
| 8 | **1.86** | **1.87** | **6.03** | **5.82** | **95.7** | **99.6** | **90.1** |
| 16 | 2.50 | 2.50 | 11.3 | 11.1 | 88.0 | 98.3 | 82.4 |

As shown in Table 11, setting $K = 4$ results in significantly worse performance across all metrics, indicating that a narrow region with too few points cannot effectively enhance query region feature representation. Conversely, $K = 16$ causes the query region to cover a broader area, reducing its ability to capture fine-grained details and produce more misleading guidance. Setting $K = 8$ strikes a balance between expressiveness and scope, achieving the best performance.

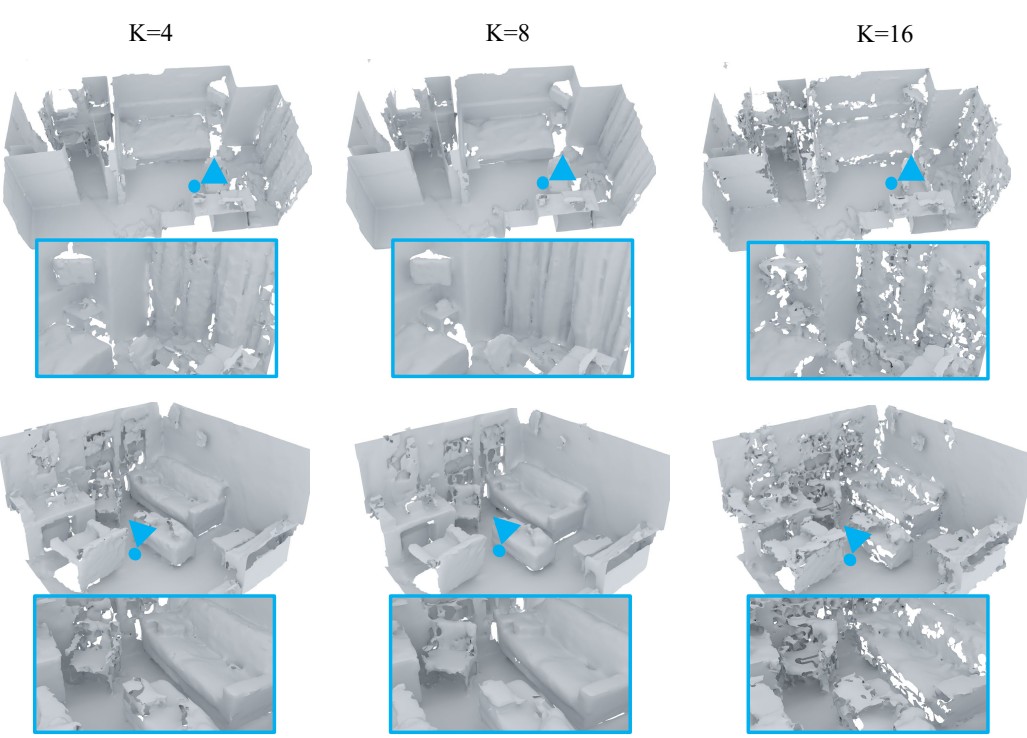

K=4      K=8      K=16

Figure 9: **KNN Size Ablation.** We selected scenes containing both simple structures, such as planes, and complex, varied geometries, highlighting the complex regions for more meaningful comparisons.

Fig. 9 illustrates the effect of $K$ on reconstruction quality. With $K = 4$, performance is adequate in planar regions but fails in complex areas (e.g., curtain undulations), causing abnormal protrusions or loss of detail. In contrast, $K = 16$ captures overly broad neighborhoods, introducing redundant information and misleading feature guidance. Instead of complementing point-wise features, it becomes detrimental. $K = 8$ provides sufficient context to capture moderately complex structures without excessively enlarging the query scope or introducing redundant information.

**Meshing Strategy**    The relative advantage of our method remains stable across both extractors. GIFS and SALS are not UDF-based and cannot be meaningfully re-meshed with UDF-specific pipelines. For UDF work ( NVF, GeoUDF, and ours) we ran controlled comparisons on ScanNet using both MeshUDF and GeoUDF (E-MC).

Table 12: **Meshing Strategy.** Report $CD_{L_1}$ and $CD_{L_2}$, F-S. with thresholds 0.005, and NC.

| Methods | $CD_{L_1} \times 10^{-3}$ | $CD_{L_2} \times 10^{-6}$ | F-S. (%) | NC (%) |
|---|---|---|---|---|
| GeoUDF (MeshUDF) | 8.96 | 89.0 | 9.6 | 79.4 |
| GeoUDF (E-MC) | 2.05 | 7.84 | 93.8 | 89.3 |
| NVF (MeshUDF) | 2.03 | 11.7 | 94.6 | 88.9 |
| NVF (E-MC) | 2.05 | 12.0 | 94.4 | 86.8 |
| Ours (MeshUDF) | 1.86 | 6.03 | 95.7 | 90.1 |
| Ours (E-MC) | **1.81** | **5.54** | **96.5** | **91.2** |

The GeoUDF paper indicates that its **E-MC method is more accurate than MeshUDF**, and our practical experience confirms this holds true in most cases. The remaining differences arise from field definitions: NVF's directions are not strictly equivalent to the UDF gradient and can exhibit directional noise near surfaces; MeshUDF's voting step partially suppresses that noise while E-MC is more sensitive to it, producing different modes. Additionally, MeshUDF is completely unsuitable for GeoUDF and will generate a double layer near the surface. We omitted E-MC for other UDF methods at 256 resolution due to its high computational cost and instead used each method's default reconstruction procedure from the original papers.

**Efficiency and Effectiveness** We report performance comparisons on ScanNet clean and noiser (added Gaussian noise with $\sigma$=0.025). Performance metrics are based on results at 256 resolution. Here we summarize the effciency metrics and measurement protocol concisely: inference time= wall-clock for the meshing pipeline (including distance inference + surface extraction); memory = peak GPU memory during inference with 100k query points.

Table 13: Comparison of methods under clean/noiser settings and efficiency metrics.

| Methods | $CD_{L_1} \times 10^{-3}$ (clean) | $CD_{L_1} \times 10^{-3}$ (noiser) | Inference time (128) | Inference time (256) | Params. (M) | Memory (G/100k pts) |
|---|---|---|---|---|---|---|
| GeoUDF | 2.05 | 10.4 | 15.3 | 86.6 | 0.775 | 7.4 |
| NVF | 2.03 | 5.20 | 2.9 | 17.0 | 10.30 | 8.9 |
| Ours | **1.86** | **4.91** | 3.1 | 18.6 | 11.37 | 20.9 |

It can be observed that although GeoUDF has the smallest number of parameters and memory footprint, it takes too long to reconstruct a room and completely fails in high-noise environments. Our approach achieves **a balance between efficiency and effectiveness**.

### A.4.3  FAILURE CASE ANALYSIS

We consider a challenging, real-world scenario: multi-tier cabinets commonly found in indoor scenes. Sparse point sampling often causes closely stacked layers to be perceived as a single surface, which blurs inter-tier boundaries and degrades reconstruction quality. In the example shown in Fig.10, all methods struggle to recover a complete, clean cabinet geometry; reconstructions exhibit substantial noise and structural collapse in the inter-tier regions.

Despite producing some local artifacts and irregular layer geometry, our method preserves clear separations between tiers: reconstructed layers show distinct gaps where the point evidence indicates separation, rather than collapsing into a single thick shell. This contrast highlights the role of our primitive construction, by encoding local structure and aggregating query-centric primitives, the method is better able to maintain layer distinctions when query points fall between surfaces. By comparison, competing methods produce largely undifferentiated or chaotic interior geometry (GIFS in this case even yields a fully enclosed volume), which demonstrates that differences in primitive/patch construction materially affect the ability to resolve multi-layer structure.

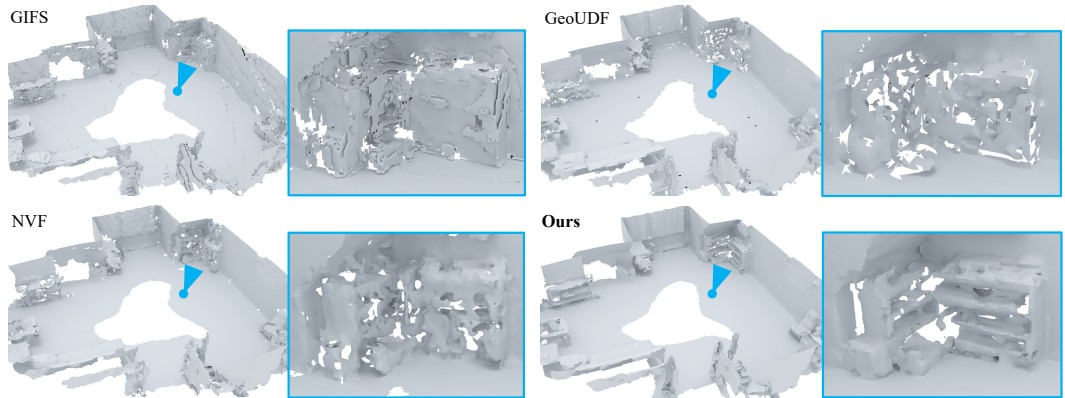

Figure 10: **Failure Case.** A typical failure case: the multi-layer cabinet. Our method preserves clear inter-layer gaps, while other approaches collapse multi-layer structures into one volume.

## A.5 OTHERS

### A.5.1 LIMITATION AND FUTURE WORK

Our RegionUDF demonstrates robust performance in reconstructing 3D scene surfaces from sparse point clouds, however, its current mesh extraction relies on adapting the MeshUDF Marching Cubes pipeline applied to a pseudo–signed distance field. A more promising approach would involve developing a dedicated algorithm for directly extracting manifolds from unsigned distance predictions, thereby avoiding the limitations of pseudo–SDF conversion. Furthermore, given that semantic labels naturally extend over continuous surfaces, integrating our implicit surface reconstruction with concomitant semantic segmentation via a unified network that predicts both geometry and semantics constitutes a compelling avenue for future research.

