# OpenReview forum: "RegionUDF: Region-Aware Unsigned Distance Fields for Surface Reconstruction from Point Clouds"
_ICLR.cc/2026/Conference — Submitted to ICLR 2026_

### Official Review · Reviewer_vLCg · 2025-10-22

**Soundness:** 3
**Presentation:** 2
**Contribution:** 3
**Rating:** 4
**Confidence:** 4

**Summary:**

The work proposes a UDF learning method from point cloud. It extracts two hierarchical features, including narrow region feature and point-wise feature, and then blend them as the local feature descriptor. The network is trained and tested on several datasets for evaluation.

**Strengths:**

The work introduces narrow, fine-grained region features to capture local details.

It decomposes the local neighborhoods into a set of triplets for primitive-based features.

**Weaknesses:**

The feature extraction of Equation (1) is not well-motivated. It'd better to explain the designing paradigm of the neural network clearly.

It seems that the K of neighboring points is important for the feature extraction. But the setting is unclear.

The trainings on different datasets are individually and the testing are also individually. Every neural network can only be used for the same category data. The generalization ability is limited.

The baseline methods are a bit out of date.

**Questions:**

$\varphi$ is defined after Equation 3, but it first appears in Equation 1.

In the first equations of Equation 1 and 2, the second $p\in N_q$, but what is the first $p$?

What is the impact of different K? K is a sensitive parameter, but it is not discussed.

How to extract surfaces from UDF?

---

> ### Author Response · Authors · 2025-11-21
> **Response to Reviewer vLCg's Comments (Part 1)**
>
> > Q1. The feature extraction of Equation (1) is not well-motivated. It'd better to explain the designing paradigm of the neural network clearly.
>
> A: The design of Equation (1) is motivated by treating each K‑NN neighborhood as an **independent mini point cloud** so the point cloud analysis techniques can be applied to learn narrow region features.
>
> This follows the **pooling+propagation paradigm** of PointNet++ (local MLP → symmetric pooling → skip/concat) and is implemented with a small MLP, mean pooling over K neighbors, and a concatenation skip connection.
>
> > Q2. It seems that the K of neighboring points is important for the feature extraction. But the setting is unclear.
>
> > Q3. What is the impact of different K? K is a sensitive parameter, but it is not discussed.
>
> A:
> The parameter $K$ in the backbone controls the receptive field size for local feature extraction. Since backbone design is not our contribution, we follow the default configuration $K=16$ and report it in the **Appendix (Parameter Settings)**.
>
> Varying K of query neighborhood changes neighborhood radius and directly affects primitive construction and final surface quality. We evaluate this K in **Appendix A.4.2 (KNN ablation)**. Empirically on our datasets:
>
> - K = 4: neighborhood too small; primitives capture overly local geometry, producing holes and under‑smoothed surfaces.
>
> - K = 8: best tradeoff; captures fine geometric detail while keeping ambiguous cross‑layer sampling low.
>
> - K = 16: neighborhood too large; ambiguity primitives increasingly mix points from adjacent layers or distant geometry, producing fragmented noise and more ambiguous structures.
>
> | K   | $CD_{L_1} \times 10^{-3}$ (Mean/Median) | $CD_{L_2} \times 10^{-6}$ (Mean/Median) | F-S. (0.005/0.01) (%) | NC (%) |
> |-----|-------|-----|---------|--------|
> | 4   | 2.00 / 2.00       | 7.40 / 7.06       | 94.2 / 99.4           | 88.6   |
> | 8   | **1.86 / 1.87**   | **6.03 / 5.82**   | **95.7 / 99.6**       | **90.1** |
> | 16  | 2.50 / 2.50       | 11.3 / 11.1       | 88.0 / 98.3           | 82.4   |
>
> As shown in **Table 11**, setting $K=4$ results in significantly worse performance across all metrics, indicating that a narrow region with too few points cannot effectively enhance query region feature representation. Conversely, $K=16$ causes the query region to cover a broader area, reducing its ability to capture fine-grained details and produce more misleading guidance due to ambiguity of primitive construction. Setting $K=8$ strikes a balance between expressiveness and scope, achieving the best performance.
>
> **Fig. 9** illustrates the effect of $K$ on reconstruction quality. With $K=4$, performance is adequate in planar regions but fails in complex areas (e.g., curtain undulations), causing abnormal protrusions or loss of detail. In contrast, $K=16$ captures overly broad neighborhoods, introducing redundant information and severe ambiguity. Instead of complementing point-wise features, it becomes detrimental, bringing numerous fragments and artifacts. $K=8$ provides sufficient context to capture moderately complex structures without excessively enlarging the query scope or introducing redundant information.

---

> ### Author Response · Authors · 2025-11-23
> **Response to Reviewer vLCg's Comments (Part 2)**
>
> > Q4. The trainings on different datasets are individually and the testing are also individually. Every neural network can only be used for the same category data. The generalization ability is limited.
>
> A: We clarify that our approach encompasses not only intra-domain evaluation but also inter-domain evaluation, demonstrating its robust generalization capabilities. We evaluate its cross-domain transfer in **Appendix A.4.1** and consistently outperform prior methods in **shape-to-scene, scene-to-shape, and scene-to-scene settings**, with both quantitative and qualitative evidence.
>
> The quantitative results **Table 9** shows NDF (Mesh) exhibits almost complete failure when generalizing from scene-level to shape-level reconstruction on Matterport3D, due to large errors in NDF predicted UDF gradients. By contrast, our method *consistently outperforms existing approaches across almost all cross-domain scenarios*, and matches their Normal Consistency when transferring from scene-level to shape-level. In the shape-to-scene setting, our model achieves performance comparable to NVF, while lower $CD_{L_2}$ and higher NC. Notably, our model **outperforms all baselines across all metrics in the scene-to-scene transfer setting**. They demonstrate that our reconstruction results more closely align with GT surfaces, and the larger $CD_{L_2}$ improvement indicates a reduction in outlier artifacts.
>
> The visual results **Fig 8** further demonstrate that our method **excels at recovering sharp and fine-grained details** in cross-domain evaluation. At the scene-to-shape level, it accurately reconstructs two closely positioned seats within a vehicle. At the shape-to-scene level, it yields noticeably sharper and more coherent object boundaries. At the scene-to-scene level, it recovers sharp structures such as wall-mounted bed curtains and crisp window-sill edges—features that competing methods fail to preserve.
>
> > Q5. The baseline methods are a bit out of date.
>
> A: We appreciate the reviewer’s concern and would like to clarify that our evaluation includes comparison with SALS (2025), a recent and competitive method for open-surface reconstruction. We reproduced SALS under identical experimental settings on both object-level and room-level benchmarks to ensure fairness.
>
> While each baseline in our manuscript has distinct strengths, our method demonstrates consistent and **often substantial improvements across nearly all settings**. For example, on watertight ShapeNet we outperform GeoUDF by *1.2% and 2.4%* in F-score, and on ABC and non-manifold ABC we surpass SALS by *1%* in F-score. On ScanNet, our method achieves a **10%** gain in $CD_{L_1}$ and nearly **20%** in $CD_{L_2}$ over NVF and GeoUDF, while on Matterport3D we obtain improvements of **5%** ($CD_{L_1}$) and **12%** ($CD_{L_2}$). Under Noise, Noiser, and Sparse ScanNet conditions, we observe F-score gains of *2–3%* over NVF and GeoUDF. We note that when F-scores exceed 90%, *even a 1% improvement is statistically meaningful*.
>
> These empirical results demonstrate that our method provides *clear advantages over both old and recent baselines*, including recent SALS.
>
> > Q6. $\varphi$ is defined after Equation 3, but it first appears in Equation 1.
>
> A: We appreciate the observation and revise the manuscript to define $\varphi$ at its first use and ensure consistent notation.
>
> > Q7. In the first equations of Equation 1 and 2, the second $p\in N_q$, but what is the first p?
>
> A: We apologize for the confusion caused by our notation. The two p refer to the same variable. We have corrected the equations and clarified the domains as:
> $r_p = \phi(p-q)\oplus (\frac{1}{K}\sum_{p}\phi(p-q))|p\in N_q, $
>
> $r_{p} = \phi(p - q)\oplus\Bigl(\tfrac{1}{3}\sum_{p}\phi(p - q)\Bigr)|p\in S, $
>
> $r_{S} = \phi(c_S)\oplus\Bigl(\tfrac{1}{K}\sum_{S}\phi(c_S - q)\Bigr)|S\in T,$
>
> > Q8. How to extract surfaces from UDF?
>
> A: Although the UDF is non-differentiable exactly on the surface, its gradients exhibit opposite signs on the two sides of the surface. This property can be exploited to recover the zero level set. In particular, UDF-based Marching Cubes *uses the dot product of gradients at two neighboring vertices to assign pseudo signs for the unsigned distances*, effectively converting them into a **pseudo–signed distance field**. This pseudo-signing step then allows the standard Marching Cubes algorithm to be applied for extracting the zero level set.
>
> We described it in **Appendix A.2.2.** As meshing is not our contribution, we adopt established strategies (MeshUDF and GeoUDF). We observed that MeshUDF can introduce artifacts at low resolutions—particularly when the segment connecting the two query points is inclined to the plane and does not intersect it—so, then, we use GeoUDF at a resolution of 128 despite its higher computational cost.

---

### Official Review · Reviewer_ZfDP · 2025-10-28

**Soundness:** 3
**Presentation:** 4
**Contribution:** 2
**Rating:** 4
**Confidence:** 4

**Summary:**

This paper proposes a region-aware unsigned distance function representation for surface reconstruction from point clouds. The key idea is to incorporate region-level and primitive-based features into the query function, aiming to enhance local geometric representation.

**Strengths:**

1. The proposed region-primitive feature construction is simple and can be easily integrated into standard UDF reconstruction pipelines.

2. The overall method is presented in a clear and structured manner, with well-defined modules and straightforward implementation steps, which makes the approach easy to follow.

**Weaknesses:**

1. The main contribution focuses on modifying feature construction within a standard UDF framework, without introducing any new geometric priors or reconstruction paradigms. This represents a relatively incremental improvement compared to recent advances such as segment-based or kernel-based reconstruction methods.

2. The paper lacks comparison with strong baselines like NKSR, which is widely recognized as a high-precision, non-learning benchmark for surface reconstruction.

3. The performance gains mainly over older UDF methods (e.g., GeoUDF, published two years ago) are modest and largely expected given the richer local feature aggregation. Without demonstrating clear advantages over stronger baselines, the contribution falls short of the level typically expected for a top-tier venue.

4. Although the motivation is to enhance local region representation, the experiments do not convincingly demonstrate qualitative superiority in challenging areas such as thin structures and boundary regions.

**Questions:**

1. The Gaussian noise level used in your experiments (σ = 0.005–0.01) appears substantially low (e.g., σ = 0.02–0.05 or with outliers). This raises concerns about the strength of the claimed robustness. Could you justify this choice of noise level, and discuss how your method would perform under more realistic or higher noise conditions? Do you expect the relative gains over existing UDF baselines to persist in such settings?



2. Your primitive-based feature uses three neighboring points to form a local patch representation. Conceptually, this seems to introduce a slightly richer local prior compared to the two-point segments used in SALS, but it is unclear why three points were specifically chosen. If the goal is to encode more complex local geometry, why not use four or more points to capture higher-order structures? Is there a principled reason for using a triplet instead of other configurations, or is this choice mainly empirical? How does the representational capacity of a triplet compare to a segment or larger patches?



3. Since the method builds on local feature enrichment rather than introducing a structural constraint, it remains unclear whether the observed improvements are robust to more challenging conditions. Could you explain why this formulation should maintain its advantages under high noise levels, sparse sampling, or real-world data, and how it provides more than incremental feature engineering?



4. A key motivation of the method is enhancing local geometric representation. However, the current experimental evidence does not clearly isolate performance in thin structures or boundary regions, which are typical failure cases for UDF methods. Could you provide targeted analysis or additional results to substantiate the claimed improvements in these challenging areas?

---

> ### Author Response · Authors · 2025-11-21
> **Response to Reviewer ZfDP's Comments (Part 1)**
>
> > Q1. The main contribution focuses on modifying feature construction within a standard UDF framework, without introducing any new geometric priors or reconstruction paradigms. This represents a relatively incremental improvement compared to recent advances such as segment-based or kernel-based reconstruction methods.
>
> A: Our method is not merely an incremental modification to the UDF framework. It provides *a distinct representational perspective* by incorporating independent local geometric information (provided solely by the neighbors of the query point) and aggregating simple primitives into complex details, thereby clearly differentiating itself from existing interpolation-based techniques and yielding substantial, demonstrable improvements.
>
> Previous methods largely overlooked *independent neighborhood geometric information* centered at each query point, relying instead on direct interpolation of features. However, such independent local cues are crucial for *capturing discriminative region patterns*. Building on this insight, we adopt a classic global–local fusion paradigm and design both *narrow region–aware aggregation and primitive-based aggregation* to aggregate simple primitives into complex details. These components effectively alleviate the limited detail-representation capacity of prior approaches and *enable the reconstruction of challenging structures*, such as multi-layer surfaces.
>
>
> > Q2. The paper lacks comparison with strong baselines like NKSR, which is widely recognized as a high-precision, non-learning benchmark for surface reconstruction.
>
> A: Thanks for the reminder. We did not include NKSR in the numerical comparisons due to concerns about potentially unfair and misleading comparisons.
>
> All baselines in our experiments are learning-based methods designed specifically for open-surface reconstruction using UDF-like fields, whereas NKSR relies on *additional geometric priors*—most notably, optimization based on the collinearity between surface gradients and normal vectors—which are *incompatible with the UDF formulation for open surfaces*. Given this *fundamental mismatch in assumptions and applicable settings*, we regard a direct quantitative comparison as inappropriate. Instead, we discuss NKSR’s key ideas in the related work section to acknowledge its contributions.
>
> > Q3. The performance gains mainly over older UDF methods (e.g., GeoUDF, published two years ago) are modest and largely expected given the richer local feature aggregation. Without demonstrating clear advantages over stronger baselines, the contribution falls short of the level typically expected for a top-tier venue.
>
> A: We appreciate the reviewer’s concern and would like to clarify that our evaluation includes comparison with SALS (2025), a recent and competitive method for learning-based open-surface reconstruction. We reproduced SALS under identical experimental settings on both object-level and room-level benchmarks to ensure fairness.
>
> While each baseline in our manuscript has distinct strengths, our method demonstrates consistent and **often substantial improvements across nearly all settings**. For example, on watertight ShapeNet we outperform GeoUDF by *1.2% and 2.4%* in F-score, and on ABC and non-manifold ABC we surpass SALS by *1%* in F-score. On ScanNet, our method achieves a **10%** gain in $CD_{L_1}$ and nearly **20%** in $CD_{L_2}$ over NVF and GeoUDF, while on Matterport3D we obtain improvements of **5%** ($CD_{L_1}$) and **12%** ($CD_{L_2}$). Under Noise, Noiser, and Sparse ScanNet conditions, we observe F-score gains of *2–3%* over NVF and GeoUDF. We note that when F-scores exceed 90%, *even a 1% improvement is statistically meaningful*.
>
> These empirical results demonstrate that our method provides *clear advantages over both old and recent baselines*, including SALS.

---

> ### Author Response · Authors · 2025-11-21
> **Response to Reviewer ZfDP's Comments (Part 2)**
>
> > Q4. The Gaussian noise level used in your experiments (σ = 0.005–0.01) appears substantially low (e.g., σ = 0.02–0.05 or with outliers). This raises concerns about the strength of the claimed robustness. Could you justify this choice of noise level, and discuss how your method would perform under more realistic or higher noise conditions? Do you expect the relative gains over existing UDF baselines to persist in such settings?
>
> A:
> - Our noise settings follow the standard protocols adopted in previous open-surface reconstruction works such as GIFS and GeoUDF: **simple shapes** (e.g., ShapeNet13) are augmented with **$\sigma$ = 0.005 noise**, while **more complex, non-watertight shapes** (e.g., ShapeNet-Cars) are evaluated **without synthetic noise**, consistent with prior practice.
>
> - While, in theory, our formulation effectively exploits *independent neighborhood geometric cues* to suppress both noise and sparsity as long as local geometry remains partially preserved. When the local geometry is heavily destroyed by extreme noise (Noiser), our formulation naturally transitions to relying more on *broader-receptive-field point-wise features via attention*, yielding a controlled and acceptable degradation. Overall, the method demonstrates **balanced robustness across multiple distribution drifts**, maintaining strong performance in both noise, sparse and cross-domain cases.
>
> - Based on the above, our comparative advantage should remain stable in challenging environments.
>
> To further verify this, we additionally evaluated our model under a much stronger noise level of $\sigma$ = **0.025** (noiser). We present results under **clean, noise, noisier, and sparse conditions** for comparison.
>
> **SALS**
> | Conditions | $CD_{L_1} × 10^{−3}$ (Mean/Median) | $CD_{L_2} × 10^{−4}$ (Mean/Median) | F-S. (0.005/0.01) (%) | NC (%) |
> |-------|--------|------|-----|--------|
> | Clean      | 2.02 / 2.02   | 0.075 / 0.069 | 94.7 / 99.3 | 87.1   |
> | Noise      | 4.80 / 4.74   | 0.370 / 0.349 | 61.4 / 91.9 | 84.0   |
> | Noiser     | 8.49 / 8.43   | 1.06 / 1.04   | 23.2 / 62.3 | 74.4   |
> | Sparse     | 4.98 / 4.92   | 0.467 / 0.426 | 61.5 / 88.9 | 83.2   |
>
> **GeoUDF**
> | Conditions | $CD_{L_1} × 10^{−3}$ (Mean/Median) | $CD_{L_2} × 10^{−4}$ (Mean/Median) | F-S. (0.005/0.01) (%) | NC (%) |
> |------------|----------------|----------------|---|--------|
> | Clean      | 2.05 / 2.04   | 0.078 / 0.075 | 93.8 / 99.0 | 89.3   |
> | Noise      | 4.03 / 4.00   | 0.240 / 0.236 | 69.8 / 95.7 | 75.5   |
> | Noiser     | 10.4 / 10.4   | 1.84 / 1.85   | 24.6 / 51.6 | 56.2   |
> | Sparse     | 2.45 / 2.42   | 0.130 / 0.122 | 90.2 / 97.5 | 87.6   |
>
> **NVF**
> | Conditions | $CD_{L_1} × 10^{−3}$ (Mean/Median) | $CD_{L_2} × 10^{−4}$ (Mean/Median) | F-S. (0.005/0.01) (%) | NC (%) |
> |------------|-------------|---------|----------|--------|
> | Clean      | 2.03 / 1.98    | 0.117 / 0.069  | 94.6 / 99.3| 88.9   |
> | Noise      | 2.88 / 2.84    | 0.171 / 0.142  | 87.2 / 97.6| 85.6   |
> | Noiser     | 5.20 / 5.23    | 0.503 / 0.469  | 62.0 / 86.9| 78.6   |
> | Sparse     | 2.77 / 2.73    | 0.199 / 0.175  | 87.2 / 96.1| 85.1   |
>
> **Ours**
> | Conditions | $CD_{L_1} × 10^{−3}$ (Mean/Median) | $CD_{L_2} × 10^{−4}$ (Mean/Median) | F-S. (0.005/0.01) (%) | NC (%) |
> |------------|--------|-------------|--------|--------|
> | Clean      | 1.86 / 1.87| 0.060 / 0.058     | 95.7 / 99.6    | 90.1   |
> | Noise      | 2.51 / 2.46| 0.112 / 0.104     | 90.9 / 98.6    | 88.4   |
> | Noiser     | 4.91 / 4.89| 0.418 / 0.412     | 64.8 / 89.0    | 80.0   |
> | Sparse     | 2.28 / 2.25| 0.125 / 0.105     | 91.6 / 98.1    | 89.4   |
>
> As shown in Tables, in $CD_{L_1}$, NVF degrades by 41.9% under Noise and 36.5% under Sparse, while our method shows smaller drops (**7% and 13.9%** less, respectively) and is *comparable* under Noiser. GeoUDF, in contrast, suffers severe instability with degeneration of 96.5% and 407% under Noise/Noiser, where our method reduces them by **61.6% and 243%**. Under Sparse, our degradation remains *close* to GeoUDF. We simultaneously evaluated SALS under these environmental conditions. The results reveal its pronounced sensitivity to geometric structure changes, where even minor shifts in data distribution can lead to more than a 100% degradation in $CD_{L_1}$ performance. These results demonstrate that our method has **balanced robustness across multiple distribution drifts**

---

> ### Author Response · Authors · 2025-11-21
> **Response to Reviewer ZfDP's Comments (Part 3)**
>
> > Q5. Your primitive-based feature uses three neighboring points to form a local patch representation. Conceptually, this seems to introduce a slightly richer local prior compared to the two-point segments used in SALS, but it is unclear why three points were specifically chosen. If the goal is to encode more complex local geometry, why not use four or more points to capture higher-order structures? Is there a principled reason for using a triplet instead of other configurations, or is this choice mainly empirical? How does the representational capacity of a triplet compare to a segment or larger patches?
>
> A:
> - To avoid confusion, we emphasize that the two‑point segment in the SALS design refers to a line segment defined by two query points, which is then used to learn whether the segment intersects the surface. By contrast, our triplet primitive is constructed from neighboring points and represents a local surface patch. Thus, while the former introduces a novel distance‑field definition, the latter serves as a fundamental component of surface representation.
>
> - The choice of three points is motivated by geometric principles: in 3D space, the simplest non-degenerate local structure is defined by three non-collinear points, which **uniquely determines a plane**.
>
> - We avoid modeling higher-order structures involving four or more points, as *complex geometries can be effectively represented through combinations of simpler planar primitives*. Directly modeling such higher-order structures can reduce generalization performance, since these complex patterns are *often highly scenario-specific* rather than broadly applicable.
>
> - Theoretically, simple 3D primitives provide one additional geometric degree of freedom than segment while avoiding the excessive expressiveness of large patches. This intermediate complexity allows them to be learned reliably with low sample complexity and then composed to approximate rich surface structures.
>
> To validate the above claims, we evaluated segment, triplet plane and 4-point patch on ScanNet, with the following results:
>
> |Methods| $CD_{L_1} × 10^{−3}$(Mean/Median)| $CD_{L_2} × 10^{−6}$(Mean/Median)|F-S.(0.005/0.01) (%) |NC (%)|
> |------------|--------|-----------|-----------|-----------|
> | Segment     | 1.96/1.95     | 6.92/6.47    |95.2/99.4|89.7|
> | Triplet Plane  | **1.86/1.87** |**6.03/5.82** |**95.7/99.6**| **90.1**|
> | Four-Point Patch   | 2.06/2.04     | 7.54/7.05 |94.3/99.3|88.3|
>
> > Q6. Since the method builds on local feature enrichment rather than introducing a structural constraint, it remains unclear whether the observed improvements are robust to more challenging conditions. Could you explain why this formulation should maintain its advantages under high noise levels, sparse sampling, or real-world data, and how it provides more than incremental feature engineering?
>
> A:
> - Section 4.4 and A.4.1 show that our method demonstrates **balanced robustness across multiple distribution drifts**, maintaining strong performance in noise, sparse and cross-domain cases.
>
> - To learning reliable pattern widely present in the scene, our formulation should achieve higher generalizability in those challenge cases. Conceptually, the robustness arises from the formulation’s incorporation of independent neighborhood information and its use of simple geometric primitives to aggregate the region representation. We designed a spherical projection method to construct primitives, minimizing the gap between primitives and the region, to promote the learning of discriminative region representations.
>   - **When local geometry is partially preserved**, our formulation constructs primitives that closely approximate the underlying regional structure. Guided by the principle that complex patterns can be composed from simpler ones, the aggregation of these primitives implicitly embeds geometric prior information. This acts as a soft geometric constraint, helping the model suppress the effects of moderate noise.
>   - **When local geometry is significantly disrupted**, the formulation naturally shifts toward relying on point-wise features aggregated over broader receptive fields through attention mechanisms. These features are less dependent on local geometric fidelity and therefore remain informative under extreme noise, providing stability when geometric cues become unreliable.
>
> - Our method goes beyond incremental feature engineering because it is guided by a principle rather than ad-hoc modifications. Inspired by the principle that complex structures can be decomposed into simpler primitives, we construct elementary triplet-based units and aggregate them, encouraging the model to learn simple yet reliable geometric patterns. These patterns enrich the region-level representation and simultaneously act as a soft geometric prior that mitigates distribution drift. This principled design demonstrates that our approach is *conceptually driven* rather than an engineering adjustment.

---

> ### Author Response · Authors · 2025-11-21
> **Response to Reviewer ZfDP's Comments (Part 4)**
>
> > Q7. Although the motivation is to enhance local region representation, the experiments do not convincingly demonstrate qualitative superiority in challenging areas such as thin structures and boundary regions.
>
> > Q8. A key motivation of the method is enhancing local geometric representation. However, the current experimental evidence does not clearly isolate performance in thin structures or boundary regions, which are typical failure cases for UDF methods. Could you provide targeted analysis or additional results to substantiate the claimed improvements in these challenging areas?
>
> A: We respectfully note that the manuscript includes some qualitative results focusing on these challenging regions. For example, **the 2nd column of Fig. 2** highlights a slender lamp arm, and **the 3rd column** shows a sharp racing wing; **the 3rd row of Fig. 3** presents a narrow boundary gap beneath an object.
>
> Specifically, we additionally disscussed **a typical failure case of multi-layer structure** in **Appendix A.4.3** of the revision: reconstructed cabinets and multi‑tier structures produced by our model retain discernible gaps between tiers, even though local artifacts appear (irregular layer geometry, slight surface contraction, or occasional partial cross‑layer connections). These artifacts are the primary failure modes, but they do not collapse distinct layers into a single undifferentiated volume as often observed in other methods.

---

### Official Review · Reviewer_iTLm · 2025-10-28

**Soundness:** 3
**Presentation:** 3
**Contribution:** 3
**Rating:** 6
**Confidence:** 3

**Summary:**

This manuscript proposes RegionUDF. a region-aware unsigned distance field (UDF) framework that improves the accuracy and robustness of open-surface reconstruction by fusing fine-grained local region features with global contextual point features. RegionUDF demonstrates good performance across synthetic and real datasets, object-level and scene-level reconstruction, and cross-domain generalization.

**Strengths:**

1. A good-written manuscript.
2. Introducing a novel and effective discriminative region representation that explicitly fuses fine-grained local neighborhood features with broad contextual point-wise features. Enhances the ability to express local geometric structures, especially for complex non-manifold structures.
3. Mitigating the over-smoothing problem inherent in point-wise features, as convincingly demonstrated by its superior qualitative results.

**Weaknesses:**

1. No comparison of optimizing speed and memory usage is provided.
2. The manuscript lacks a discussion of failure cases.
3. UODF (CVPR’24) should be included in comparison methods.

**Questions:**

1. UODF (Unsigned Orthogonal Distance Fields: An Accurate Neural Implicit Representation for Diverse 3D Shapes, CVPR’24), which performs well on watertight shapes, should be included in comparison methods.
2. Optimizing speed, memory usage, and model parameters should to be reported and compared with main baselines.
3. In Tab.8 in Suppl., why gradient loss seems less critical for RegionUDF? Only depends on the introduction of region features?
4. Will the proposed method fail on multi-layered or complex topological structures? I suggest authors report the failure cases for discussion.

---

> ### Author Response · Authors · 2025-11-21
> **Response to Reviewer iTLm's Comments**
>
> > Q1. No comparison of optimizing speed and memory usage is provided.  	Optimizing speed, memory usage, and model parameters should to be reported and compared with main baselines.
>
> A: Thank you for your suggestion. We report performance comparisons on ScanNet clean, noise (added Gaussian noise with $\sigma=0.005$) and sparse (1/3 input points) conditions. Geometric metric is based on results at 256 resolution. Here we summarize the effciency metrics and measurement protocol concisely: inference time= wall‑clock for the meshing pipeline (including distance inference and surface extraction); memory = peak GPU memory during inference with 100k query points.
> We denote $d_C=CD_{L_1}\times 10^3$.
> |Methods|$d_C$(clean)|$d_C$(noise)|$d_C$(sparse) |Inference time (128)(s)|Inference time (256)(s)|Params. (M) |Memory (G/100k pts)|
> |------------|-----------|-----------|-----------|-----------|-----------|-----------|-----------|
> | GeoUDF     |2.05 |4.03|2.45|15.3|86.6 |0.775 |7.4|
> | NVF     | 2.03|2.88|2.77|2.9|17.0 |10.30 |8.9|
> | Ours     | 1.86|2.51|2.28|3.1|18.6 |11.37 |20.9|
>
> It can be observed that although GeoUDF has the smallest number of parameters and memory footprint, it takes too long to reconstruct a room and completely fails in noise condition, and NVF degrades significantly in sparse condition. Therefore, our approach achieves **a balance between efficiency and effectiveness**.
>
> > Q2. The manuscript lacks a discussion of failure cases. 	Will the proposed method fail on multi-layered or complex topological structures? I suggest authors report the failure cases for discussion.
>
> A: We acknowledge that multi‑layered geometry and complex topologies remain challenging for our method, but our method more reliably **preserves separations between adjacent layers** than competing approaches.
>
> We added a dedicated failure‑case analysis to the Appendix (**see A.4.3 and Fig.10**): reconstructed cabinets and multi‑tier structures produced by our model retain **discernible gaps between tiers**, even though local artifacts appear (irregular layer geometry, slight surface contraction, or occasional partial cross‑layer connections). These artifacts are the primary failure modes, but our method *do not collapse distinct layers into a single undifferentiated volume* as often observed in other methods.
>
> > Q3. UODF (CVPR’24) should be included in comparison methods. 	UODF (Unsigned Orthogonal Distance Fields: An Accurate Neural Implicit Representation for Diverse 3D Shapes, CVPR’24), which performs well on watertight shapes, should be included in comparison methods.
>
> A: Thank you for the reminder. We have described UODF in *the related work section (lines 114–117)*, noting its definition of minimal unsigned distance along three orthogonal directions, which allows each spatial point to directly access its closest surface point and thereby achieve high‑precision reconstruction.
>
> Regarding the quantitative comparison, we decided not to include UODF results because *its evaluation protocol differs from ours*: our method is trained on one batch of data and tested on another, whereas UODF follows a per‑scene optimization strategy. *To avoid potentially unfair or misleading comparisons under these differing settings*, we chose to highlight UODF in the related work instead of reporting numerical results side by side.
>
> > Q4. In Tab.8 in Suppl., why gradient loss seems less critical for RegionUDF? Only depends on the introduction of region features?
>
> A: Our region-aware features provide stronger representational capacity than alternative designs, allowing the model to more effectively learn an accurate distance field. Because the gradient field is directly derived from the distance field, *a well‑captured distance field, enabled by these enriched features, ensures that gradients can be reliably obtained without heavy explicit supervision.* In this way, the improved representational power naturally **reduces dependence on gradient supervision** and **diminishes its marginal utility**.
>
> Yes, this depends on the introduction of our region-aware features. To substantiate this argument, we additionally performed ablation experiments on the $C_1$ variant (weaker than $C_2$):
>
> We denote $d_C=CD_{L_2} × 10^{6}$, F-S. (%) with threshold 0.005 and NC (%).
> | Variants | $\lambda_2=0$ ($d_C$/F-S./NC) | $\lambda_2=1e^{-2}$ ($d_C$/F-S./NC) | $\lambda_2=1e^{-3}$ ($d_C$/F-S./NC) | $\lambda_2=1e^{-4}$ ($d_C$/F-S./NC) |
> |---------|:------:|:------:|:-------:|:---------:|
> | C₁      | 7.56 / 94.9 / 89.4 | 9.52 / 92.7 / 88.0       | **6.33 / 95.3 / 89.8**   | 7.71 / 94.0 / 88.7       |
> | C₂      | 6.16 / 95.5 / 89.7 | 6.87 / 94.9 / 89.3       | **6.03 / 95.7 / 90.1**   | 6.24 / 95.4 / 89.8       |
>
> $C_1$ exhibits relatively weaker feature representation capabilities than $C_2$, so it **relies more heavily** on gradient supervision to converge to a better distance field.

---

### Official Review · Reviewer_4PSj · 2025-10-31

**Soundness:** 3
**Presentation:** 3
**Contribution:** 1
**Rating:** 4
**Confidence:** 3

**Summary:**

RegionUDF proposes a region-aware UDF for surface reconstruction that fuses point-wise contextual features with local region features extracted from the K-NN neighborhood of each query. The method also introduces a primitive-based decomposition (angle-sorted triplets on a projected sphere) that builds primitive features which are aggregated into the final query feature.

**Strengths:**

One major technical contribution is the combination of wide-context point-wise features with narrow region features that addresses a weakness of point-based UDF methods (oversmoothing vs. noisy local detail). The primitive decomposition is a plausible way to capture multi-patch local structure.

The authors evaluate on watertight ShapeNet, ShapeNet Cars (open), ABC / non-manifold ABC, ScanNet and Matterport3D (room-scale). They report improvements across object and scene scales (tables / figures).

**Weaknesses:**

The authors still rely on MeshUDF / GeoUDF meshing heuristics and remove vertices beyond a threshold. They note this as a limitation and suggest a dedicated UDF->manifold extractor as future work. This is important because some reconstruction improvements may come from meshing heuristics rather than the learned field itself. Please be explicit about how much improvement remains when using the same meshing pipeline for all methods.

Are the improvements stable across different meshing heuristics (MeshUDF vs GeoUDF) when they are applied identically to all methods? Please report both sets of numbers or at least clarify.

In the ablation study, the gains could partially stem from a stronger backbone (PointTransformer V2) or network capacity. The ablation compares to POCO-style baselines but I’d like a controlled experiment: same backbone capacity, with and without the region/primitive module. It would be better to show parameter counts and runtime to rule out trivial capacity explanations.

L470: “Abaltion” → “Ablation” appears in the appendix headings

**Questions:**

The primitive construction uses spherical projection and angle-sorting. The paper acknowledges ambiguity in multi-layered structures (Sec A.2), but does not quantify failure cases. How often primitives span different surfaces, and whether that harms the result?

You report training times on 2×RTX3090 (56h / 24h). What is the inference time to reconstruct a single room (including meshing) at resolution 128 and 256? Peak GPU memory?

---

> ### Author Response · Authors · 2025-11-21
> **Response to Reviewer 4PSj's Comments (Part 1)**
>
> > Q1. The authors still rely on MeshUDF / GeoUDF meshing heuristics and remove vertices beyond a threshold. They note this as a limitation and suggest a dedicated UDF->manifold extractor as future work. This is important because some reconstruction improvements may come from meshing heuristics rather than the learned field itself. Please be explicit about how much improvement remains when using the same meshing pipeline for all methods.
>
> > Q2. Are the improvements stable across different meshing heuristics (MeshUDF vs GeoUDF) when they are applied identically to all methods? Please report both sets of numbers or at least clarify.
>
> A: Thanks for your valuable comment. The relative advantage of our method remains stable across both extractors.
>
> GIFS and SALS are not UDF-based and cannot be meaningfully re-meshed with UDF-specific pipelines. For UDF work (NVF, GeoUDF, and ours) we ran controlled comparisons on ScanNet using both MeshUDF and GeoUDF (E‑MC).
>
> |Methods| $CD_{L_1} × 10^{−3}$| $CD_{L_2} × 10^{−6}$| F-S.(0.005) (%) | NC (%)|
> |------------|--------|-----------|-----------|-----------|
> | GeoUDF  (MeshUDF)   | 8.96     | 89.0     |9.6|79.4|
> | GeoUDF   (E-MC)  | 2.05     | 7.84     |93.8|89.3|
> | NVF  (MeshUDF) | 2.03     | 11.7 |94.6|88.9|
> | NVF  (E-MC) | 2.05     | 12.0 |94.4|86.8|
> | Ours (MeshUDF)   | 1.86     | 6.03|95.7|90.1|
> | Ours (E-MC)   | 1.81     | 5.54 |96.5|91.2|
>
> The GeoUDF paper indicates that its **E-MC method is more accurate than MeshUDF**, and our practical experience confirms this holds true in most cases. The remaining differences arise from field definitions: NVF’s directions are not strictly equivalent to the UDF gradient and can exhibit directional noise near surfaces; MeshUDF’s voting step partially suppresses that noise while E‑MC is more sensitive to it, producing different modes. Additionally, MeshUDF is completely unsuitable for GeoUDF and will generate a double layer near the surface.
>
>
> > Q3. In the ablation study, the gains could partially stem from a stronger backbone (PointTransformer V2) or network capacity. The ablation compares to POCO-style baselines but I’d like a controlled experiment: same backbone capacity, with and without the region/primitive module. It would be better to show parameter counts and runtime to rule out trivial capacity explanations.
>
> A: Thank you for this important point. We would claim all variants ($B_1, B_2, C_0, C_1, C_2$) in Table 5 are controlled experiments that share the *exact same backbone (PointTransformer V2)* and *similar network capacity*; the only differences are the small modules designed to produce query region features.
>
> We denote $C_0$ as the pure interpolation baseline (Eq. 4), $C_1$ as the narrow region‑aware aggregation architecture (Eq. 1, Sec. 3.1), and $C_2$ as the primitive‑based architecture (Eqs. 2–3). We further denote $B_1$ as the variant where positional‑encoding interpolation (Eq. 5) replaces the pure interpolation, and $B_2$ as the variant where AttSet replaces the attention module in $B_1$.
>
> Concretely, $C_0$→$C_1$ isolates the **narrow region-aware aggregation vs. pure interpolation**, $C_1$→$C_2$ isolates the **contribution of primitive-based aggregation**, $C_1$ vs $B_1$/$B_2$ isolates the **region‑aware design versus positional‑encoding interpolation**, and $B_1$→$B_2$ **isolates and eliminates the influence of the attention module**. Additionally, we update the number of parameters for each model as follows:
>
> | Variants | $CD_{L_1} \times 10^{-3}$ (Mean/Median) | $CD_{L_2} \times 10^{-6}$ (Mean/Median) | F-S.(0.005/0.01) (%) | NC (%) | Params. (M) |
> |---------|:------------:|:------------:|:--------:|:--------:|:------:|
> | $B_1$| 1.92/1.91 | 6.33/6.02 | 95.3/99.5| 89.0   | 11.32|
> | $B_2$| 1.92/1.91 | 6.30/6.00 | 95.3/99.5| 89.2   | 11.36|
> | $C_0$| 2.10/2.07 | 8.20/7.73 | 93.7/99.1| 88.6   | 11.37|
> | $C_1$| 1.90/1.90 | 6.33/6.06 | 95.3/99.5| 89.8   | 11.36|
> | $C_2$| **1.86/1.87**   | **6.03/5.82**   | **95.7/99.6** | **90.1** | 11.37|
>
> As shown in Table, the metrics for $C_0$, $C_1$, and $C_2$ exhibit a clear upward trend. $C_0$ serves as a baseline, whereas $C_1$ significantly surpasses it, demonstrating the **benefit of narrow region features**. $C_2$ further improves upon $C_1$, indicating that **primitive-based patch feature extraction effectively captures finer surface details**. Comparisons between $B_1$ and $B_2$ reveal **minimal impact from the AttSet module**, while $C_1$ outperforms both, highlighting that **our region-aware design is more expressive than positional‑encoding interpolation**.

---

> ### Author Response · Authors · 2025-11-21
> **Response to Reviewer 4PSj's Comments (Part 2)**
>
> > Q4. The primitive construction uses spherical projection and angle-sorting. The paper acknowledges ambiguity in multi-layered structures (Sec A.2), but does not quantify failure cases. How often primitives span different surfaces, and whether that harms the result?
>
> A: The frequency with which a primitive spans multiple surfaces primarily *depends on the relative position of the query point*. When the query point lies **outside** a multi-layer structure, spherical projection cannot distinguish between layers and ambiguity arises; when the query point lies **between** layers, the two sides become separable, and the primitives can be reliably assigned.
>
> This ambiguity *can reduce the reliability* of individual primitives, but its effect on the final reconstruction **is limited** and remains *smaller* than the degradation caused by Euclidean distance–based partitioning.
>
> - When the query point is **outside** two layers, the primitives may treat them as a single surface, yet this limitation **is not unique** to our method and also appears in Euclidean distance–based partitioning.
>
> - When the query point lies **between** layers, our formulation can **effectively separate** them, which Euclidean distance–based method cannot achieve.
>
> - Moreover, during primitive aggregation, the attention mechanism operates over primitive centers, enabling the model to reinterpret ambiguous primitives as representing the “in-between” region of two layers. These primitives are then **compensated by** those constructed from query points located between the layers, which provide clear and consistent cues.
>
> Thus, ambiguity may introduce minor artifacts, such as closer surfaces or a few unintended connections, but *overall the method retains clear advantages* over Euclidean distance–based partitioning, as demonstrated by the resulting reconstructions on ScanNet:
>
> |Methods| $CD_{L_1} × 10^{−3}$(Mean/Median)| $CD_{L_2} × 10^{−6}$(Mean/Median)|F-S.(0.005/0.01) (%) |NC (%)|
> |------------|--------|-----------|-----------|-----------|
> | Euclidean distance-based   | 2.11/2.11 | 8.43/8.21 |93.1/99.1|88.6|
> | Spherical projection     | 1.86/1.87| 6.03/5.82 |95.7/99.6 |90.1|
>
> Alternatively, although our approach exhibits certain ambiguities and partial failures in reconstructing multi‑layered structures, it is still able to approximate and **preserve the intervals between layers**. As shown in **Appendix A.4.3 and Figure 10**, even when the reconstructed layers contain *minor artifacts or occasional cross‑layer connections*, the overall geometry maintains **clear separation between tiers**—contrasting with other methods that collapse the layers entirely and fail to achieve stratification.
>
> > Q5. You report training times on 2×RTX3090 (56h / 24h). What is the inference time to reconstruct a single room (including meshing) at resolution 128 and 256? Peak GPU memory?
>
> A: We report performance comparisons on ScanNet clean, noise (added Gaussian noise with $\sigma=0.005$) and sparse (1/3 input points) conditions. Geometric metric is based on results at 256 resolution. Inference time denotes the avg time to reconstruct one room. Memory denotes peak GPU memory during inference with 100k query points. We denote $d_C=CD_{L_1}\times 10^3$.
> |Methods|$d_C$(clean)|$d_C$(noise)|$d_C$(sparse) |Inference time (128)(s)|Inference time (256)(s)|Params. (M) |Memory (G/100k pts)|
> |------------|-----------|-----------|-----------|-----------|-----------|-----------|-----------|
> | GeoUDF     |2.05 |4.03|2.45|15.3|86.6 |0.775 |7.4|
> | NVF     | 2.03|2.88|2.77|2.9|17.0 |10.30 |8.9|
> | Ours     | 1.86|2.51|2.28|3.1|18.6 |11.37 |20.9|
>
> It can be observed that although GeoUDF has the smallest number of parameters and memory footprint, it takes too long to reconstruct a room and completely fails in noise condition, and NVF degrades significantly in sparse condition. Therefore, our approach achieves **a balance between efficiency and effectiveness**.
>
> > Q6. L470: “Abaltion” → “Ablation” appears in the appendix headings
>
> A: Thank you for your correction. We have carefully reviewed similar typos and addressed them in the revision. Specifically, in Table 1 the header “Midian” has been corrected to “Median.” In Tables 6, the notation $CD_{L1}$ and $CD_{L2}$ has been standardized to $CD_{L_1}$ and $CD_{L_2}$ to ensure consistency with other symbols. In addition, the index notation “i → j” in line 173 has been corrected.

---

### Author Response · Authors · 2025-11-27
**Rebuttal and Revision Roadmap**

We conducted a comprehensive revision of the entire manuscript to fully incorporate the constructive feedback from all reviewers, with all changes highlighted in blue. Below, we provide a roadmap that links each reviewer's concerns to the corresponding rebuttal and revised sections in the manuscript.

We denote 'Q' as weakness or question. We denote 'A' as Answer. 'A. x' represents Appendix section, while 'Sec. x' represents Main Text section.

----
**Reviewer 4PSj**

- Q1–Q2 (meshing strategy): [A1] We added ablation studies with alternative meshing strategies to exclude meshing effects (A.4.2, Table 12).

- Q3 (backbone): [A2] We reinterpreted module ablations (all of them had the same backbone) to show that network capacity did not account for the observed gains (Sec. 4.4, Table 5).

- Q4 (primitive limitation): [A3] We analyzed conditions that trigger the primitive‑construction limitation and demonstrated its limited impact on final reconstructions with additional experiments (A.2; Sec. 4.4, Table 7).

- Q5 (efficiency): [A4] We reported a comprehensive efficiency–effectiveness comparison against baselines, showing a favorable trade‑off (A.4.2, Table 13).

- Q6 (notation/typos): [A5] We standardized notation in Equations 1–3, corrected typographical errors, and ensured symbol consistency.

----
**Reviewer iTLm**

- Q1 (efficiency): [A1] We reported a comprehensive efficiency–effectiveness comparison against baselines, showing a favorable trade‑off (A.4.2, Table 13).

- Q2 (failure cases): [A2] We acknowledged remaining challenges for complex topologies and added a canonical failure case (multi‑layer structures), showing that our method better recovers inter‑layer gaps than baselines (A.4.3).

- Q3 (baseline selection): [A3] We explained concerns about unfair comparisons and therefore excluded UODF from quantitative and qualitative comparisons.

- Q4 (gradient supervision): [A4] We showed that region‑aware features reduced dependence on gradient supervision and supported this with ablation results (A.4.2, Table 10).

----
**Reviewer ZfDP**

- Q1 (contribution): [A1] We clarified that our core contribution is representing independent geometric substructures as primitives, grounded in the compositional principle that complex shapes decompose into simple primitives (Sec. 1; Sec. 3.2; A.2).

- Q2 (baseline selection): [A2] We justified excluding NKSR from comparisons due to fairness concerns.

- Q3 (performance): [A3] We reported significant performance gains, including comparisons to the latest SALS (2025).

- Q4 (robustness): [A4] We justified our noise selection, added the noiser condition experiment and demonstrated balanced robustness across diverse distributional shifts (Sec. 4.4, Table 6).

- Q5 (primitive construction): [A5] We clarified the principle of primitive construction. We further provided conceptual comparisons between triplet primitives and alternatives (segments, higher‑order patches) and validated the design experimentally (Sec. 4.4, Table 8).

- Q6 (robustness of methodology): [A6] We analyzed how explicit neighborhood structure supplies robust guidance and empirically verified improved robustness to distribution drift (Sec. 4.4, Table 6).

- Q7–Q8 (challenging cases): [A7] We clarified that visualizations already included detailed comparisons and added a new failure‑case comparison to further illustrate improved detail representation (A.4.3).

----
**Reviewer vLCg**

- Q1 (motivation of Eq. 1): [A1] We clarified that Eq. 1 follows the pooling–propagation paradigm of PointNet++.

- Q2–Q3 (K for KNN): [A2] We reinterpreted the effect of K on region‑aware features and reported sensitivity analyses (A.4.2, Table 11; Fig. 9).

- Q4 (generalizability): [A3] We clarified that we have already reported cross‑domain validation to demonstrate generalizability (A.4.1).

- Q5 (baseline selection): [A4] We clarified that comparisons included the latest SALS (2025).

- Q6–Q7 (notation/typos): [A5] We standardized notation in Equations 1–3 and corrected spelling and symbol inconsistencies.

- Q8 (meshing strategy): [A6] We clarified that meshing strategy for UDFs was discussed in A.2.2.

---

### Author Response · Authors · 2025-12-03
**Author Remarks to the Area Chair**

We sincerely thank the Area Chair for the time and effort devoted.

To facilitate your final assessment and minimize your workload, we provide a recap of our paper's core storyline, a summary of reviews and responses, and a guide to our revisions.

---
### **Core Storyline**

We reveal that classical interpolation schemes used in UDF learning induce an **oversmoothing** effect and **insufficient local‑detail representation**: weighted pooling of query‑point features attenuates high‑frequency neighborhood signals and mixes information from distinct surface layers, which degrades the representation of sharp edges and other local details.
We investigated how underlying neighborhood structure affects local detail representation. Motivated by the compositional principle that *complex structures decompose into simple primitives*, points flanking a query located between multiple layers can be partitioned into regional groups and further subdivided into primitives, thereby providing explicit structural guidance for local detail representation.
Based on this insight, we designed **RegionUDF** to preserve the underlying structure of query point neighborhoods for more refined local detail representation.
RegionUDF achieves state‑of‑the‑art performance on shape‑level and room‑level open surface reconstruction benchmarks, substantially improves recovery of sharp edges and multi‑layer structures, and demonstrates robustness to distributional shifts and superior generalization across diverse scenes.

---
### **Summary of Reviews and Responses**

In initial reviews, the scores are 4,6,4,4. The reviewers recognized and highlighted:

- **Simplicity and integrability** of the region‑primitive feature construction facilitate its adoption in standard UDF pipelines.
- **Clarity, structural organization, and straightforwardness** of the theoretical description and implementation make the approach easy to follow.
- **Superior empirical results** at both object and scene scales, demonstrated through quantitative metrics and qualitative visualizations.
- **Soundness for identifying multi‑layer structures**, demonstrated by intuitive analysis and superior qualitative visualizations.

In the rebuttal, we resolved common concerns by:

- **Conducting deeper methodological investigation**: We performed a conceptual analysis comparing the adopted triplet‑primitive construction with alternative local representations such as segments and higher‑order patches, and we presented experimental evidence demonstrating the triplet primitive’s advantages. We additionally analyzed how underlying structures of query‑point neighborhoods supplied supplementary guidance, experimentally verifying that this increased robustness to distributional shifts.
- **Improving the presentation and exposition**: We further clarified the motivation, explicitly enumerated our contributions, standardized symbol representations and corrected typos. We also reorganized and rewrote the analyses accompanying the key tables to improve clarity and better substantiate our claims. Finally, we clarified the baseline selection criteria and rationale to justify the comparative evaluations.
- **Completing all requested empirical investigations with clear analyses**: Following the reviewers' suggestions, we **added full set of additional experiments and provided corresponding analyses**, including validation of the primitive‑construction principle, control experiments excluding meshing‑strategy effects, control experiments excluding network capacity effects, evaluations of efficiency and effectiveness, studies of region‑aware feature dependence on gradient supervision, and robustness tests under increased noise. These results collectively reinforced the robustness, theoretical soundness, and practical relevance of RegionUDF.

**One reviewer (zfDP) raised his score from 4 to 6** by Nov. 27. We **have carefully addressed all their comments**, and **have incorporated the corresponding revisions into the revised manuscript**. We found their questions and suggestions highly constructive, and we believe our revisions have significantly improved the clarity, scientific significance of the paper.

---

### Meta-Review · Area_Chair_tC3k · 2025-12-23

**Summary:**

The reviewers acknowledged that RegionUDF is a sound and well-engineered method that improves local detail representation in UDF-based surface reconstruction through region-aware and primitive-based feature aggregation. However, concerns were raised about the incremental nature of the contribution, baseline selection and fairness, dependence on meshing heuristics, and the need for clearer motivation, efficiency analysis, and failure-case discussion. Overall, the paper was viewed as technically solid but borderline in novelty.

**Reviewer Concerns:**

**Addressed by the rebuttal:**

Clarified the core contribution and motivation of region-primitive construction.

Provided controlled ablations ruling out backbone capacity and meshing-strategy effects.

Added efficiency, memory, and inference-time analysis.

Included robustness experiments under noise, sparsity, and cross-domain settings.

Added failure-case analysis for multi-layer and complex structures.

Justified baseline inclusion/exclusion (e.g., NKSR, UODF) and added comparisons with recent methods (e.g., SALS 2025).

Improved presentation, notation, and experimental clarity.

**Still outstanding:**

Some reviewers may still view the method as incremental feature engineering rather than a paradigm shift.

Comparisons with non-learning or fundamentally different reconstruction paradigms remain limited by fairness concerns.

Improvements on thin structures  could be further isolated and quantified.

**Reviewer Scores:**

Reviewer 4PSj: Likely slightly more positive.

Reviewer iTLm: Likely unchanged or slightly improved.

Reviewer ZfDP: Likely improved marginally.

Reviewer vLCg: Likely unchanged or marginally improved.

---

### Decision · Program_Chairs · 2026-01-26

Reject